# Multiple pairs of allelic MLA immune receptor-powdery mildew AVR<sub>A</sub> effectors argue for a direct recognition mechanism

Isabel ML Saur[1], Saskia Bauer[1], Barbara Kracher[1], Xunli Lu[1†], Lamprinos Franzeskakis[2], Marion C Müller[3], Björn Sabelleck[2], Florian Kümmel[2], Ralph Panstruga[2], Takaki Maekawa[1], Paul Schulze-Lefert[1,4]*

[1]Department of Plant Microbe Interactions, Max Planck Institute for Plant Breeding Research, Cologne, Germany; [2]Unit of Plant Molecular Cell Biology, Institute for Biology I, RWTH Aachen University, Aachen, Germany; [3]Department of Plant and Microbial Biology, University of Zurich, Zurich, Switzerland; [4]Cluster of Excellence on Plant Sciences, Düsseldorf, Germany

**Abstract** Nucleotide-binding domain and leucine-rich repeat (NLR)-containing proteins in plants and animals mediate intracellular pathogen sensing. Plant NLRs typically detect strain-specific pathogen effectors and trigger immune responses often linked to localized host cell death. The barley *Mla* disease resistance locus has undergone extensive functional diversification in the host population and encodes numerous allelic NLRs each detecting a matching isolate-specific avirulence effector (AVR<sub>A</sub>) of the fungal pathogen *Blumeria graminis* f. sp. *hordei* (*Bgh*). We report here the isolation of *Bgh* AVR<sub>a7</sub>, AVR<sub>a9</sub>, AVR<sub>a10</sub>, and AVR<sub>a22</sub>, which encode small secreted proteins recognized by allelic MLA7, MLA9, MLA10, and MLA22 receptors, respectively. These effectors are sequence-unrelated, except for allelic *AVR<sub>a10</sub>* and *AVR<sub>a22</sub>* that are co-maintained in pathogen populations in the form of a balanced polymorphism. Contrary to numerous examples of indirect recognition of bacterial effectors by plant NLRs, co-expression experiments with matching *Mla-AVR<sub>a</sub>* pairs indicate direct detection of the sequence-unrelated fungal effectors by MLA receptors.
DOI: https://doi.org/10.7554/eLife.44471.001

*For correspondence:
schlef@mpipz.mpg.de

Present address: †Department of Plant Pathology, College of Plant Protection, China Agricultural University, Beijing, China

Competing interests: The authors declare that no competing interests exist.

## Introduction

The NLR family of immune receptors is structurally conserved between animals and plants, perceives non-self and modified-self molecules inside host cells, and mounts potent innate immune responses to terminate microbial pathogenesis (*Maekawa et al., 2011a*). Animal NLRs are normally activated by conserved microbe- or damage-associated molecular patterns (MAMPs/DAMPs), whereas plant NLRs typically detect strain-specific pathogen effectors, designated avirulence effector proteins (AVRs) (*Maekawa et al., 2011a*; *Jones et al., 2016*). Plant NLRs recognize either the effector structure or sense effector-mediated modifications of host proteins (*Jones et al., 2016*). Single plant NLRs or NLR pairs can also detect the manipulation of effector target mimics, called decoys, and a variation of the latter mechanism involves the direct integration of decoy domains mimicking host targets within NLRs, termed integrated decoy domains (*Jones et al., 2016*; *Cesari et al., 2014*; *Kroj et al., 2016*). The selective forces shaping the evolution of the comparatively small complement of NLRs in vertebrates are incompletely understood (~20 family members, (*Lange et al., 2011*; *Meunier and Broz, 2017*)), but in plant species co-evolution with host-adapted pathogens has strongly influenced the expansion and diversification of NLR repertoires (*Jacob et al., 2013*; *Meyers et al., 2005*).

**eLife digest** Powdery mildews are fungal diseases that affect many plants, including important crops such as barley. The fungi behind these diseases deliver molecules known as effectors inside plant cells, which manipulate the plants' biology and help the fungus to invade the plants' tissues. In response, some plants have evolved immune receptors encoded by so-called *R* genes (short for resistance genes) that detect the effectors inside the plant cell and trigger an immune response. The response often kills the plant cell and those nearby to limit the spread of the fungus. Effectors that are recognized by host immune receptors are termed avirulence effectors (or AVRs for short).

Scientists tend to assume that most effectors do not bind directly to their immune receptors. Instead, it is thought that the immune receptors are more likely to be detecting a change in some other plant protein that is caused by the effectors' activities.

In barley populations, one *R* gene that protects against powdery mildew encodes an immune receptor known as MLA. Different plants can carry subtly different versions of this *R* gene meaning that they make similar but different variants of the same receptor. Each MLA variant confers immunity only to strains of powdery mildew that carry the matching AVR effector. A few AVR effectors from powdery mildews have been identified, but most AVR effectors from powdery mildews remain unknown.

Saur et al. looked for new AVR effectors from powdery mildew fungi collected in the field, and found four that were recognized by barley plants carrying MLA variants. Two of these new effectors were fairly similar to each other, but they were all unlike those that had been identified previously.

When Saur et al. engineered barley cells to make these new AVRs alongside their matching MLA receptors, the cells died – which is consistent with the expected immune response. Similar experiments with distantly related tobacco plants agave the same results. This suggested that the immune receptors did not need any other barley proteins to recognize the effectors, indicating that the interaction between the two may be direct. Indeed, two other techniques that test for direct protein-protein interactions, – one that involved extracts from tobacco leaves, and another that involved yeast, – gave results consistent with a direct interaction between the MLA receptor variants and the fungal effectors.

Plant disease is still a major cause of loss of yield in crops. Transferring an *R* gene from one plant species to another is a potentially powerful approach to help crops resist disease. The discovery that multiple variants of the same resistance gene can bind to dissimilar effectors from a disease-causing fungus in distantly related plant species underlines the potential of this approach.

DOI: https://doi.org/10.7554/eLife.44471.002

Plant disease resistance (*R*) genes to host-adapted pathogens often encode NLRs, are frequently members of large gene families organized in complex clusters of paralogous genes, and can rapidly evolve through a range of natural gene diversification mechanisms (*Jacob et al., 2013*; *Meyers et al., 2005*). There are several examples of allelic series of NLR-type *R* genes known in plants (*Ellis et al., 1999*; *Allen et al., 2004*; *Srichumpa et al., 2005*; *Seeholzer et al., 2010*; *Kanzaki et al., 2012*). In these cases, multiple distinct recognition specificities evolved in the host population at a single *R* gene with each allele detecting a corresponding strain-specific AVR in the pathogen population. Such multi-allelic NLR-type *R* genes are particularly instructive for studying the underlying co-evolutionary process between host and pathogen.

Ascomycete powdery mildews are widespread pathogens of thousands of angiosperm plant species in temperate climates, including economically relevant crops (*Glawe, 2008*). They are obligate biotrophic pathogens, meaning that their growth and reproduction is entirely dependent on living host cells. The haploid barley powdery mildew pathogen *Blumeria graminis* forma specialis *hordei* (*Bgh*) multiplies mainly clonally and is a member of the species *Blumeria graminis* that is specialized for its host plant barley (*Hordeum vulgare*). There are various specialized forms (*formae speciales* or fs. spp.) of *B. graminis*, each of which is capable of infecting the respective host plant species belonging to the grass (Poaceae) family, including cereals such as barley and wheat (*Wyand and Brown, 2003*). Within each *forma specialis*, numerous isolates (strains) can be distinguished in the pathogen population, based on their respective virulence or avirulence infection phenotypes vis-à-

vis particular genotypes of the host population (*Lu et al., 2016*). The genomes of powdery mildews are characterized by the loss of several, otherwise widely conserved Ascomycete genes with functions related to carbohydrate degradation and primary and secondary metabolism (*Spanu et al., 2010*; *Wicker et al., 2013*), and this is believed to explain their strict dependence on living plant cells. Similar to other filamentous phytopathogens, grass-infecting powdery mildew genomes harbor hundreds of candidate secreted effector protein (CSEP)-coding genes, which are assumed to contribute to fungal pathogenesis (*Wicker et al., 2013*; *Pedersen et al., 2012*; *Guttman et al., 2014*). Pathogen effectors often work by subverting innate immune responses, thereby facilitating host colonization and disease (*Rovenich et al., 2014*).

Domesticated barley and wheat contain numerous powdery mildew *R* gene loci that were often introgressed from their corresponding wild relatives (*Jørgensen and Wolfe, 1994*; *Lutz et al., 1995*; *Maekawa et al., 2019*). In both barley and its close relative wheat, one of these powdery mildew *R* loci, designated *mildew locus a* (*Mla*) and *powdery mildew 3* (*Pm3*), respectively, has been subject to exceptional functional diversification, resulting in large numbers of *Mla* or *Pm3* recognition specificities (*Seeholzer et al., 2010*; *Lutz et al., 1995*; *Bhullar et al., 2010*). Although wheat *Pm3* and barley *Mla* loci each span a cluster of *NLR* genes, known *Pm3* and *Mla* recognition specificities to the *B. graminis* pathogen appear to have arisen from allelic diversification of a single *NLR* gene in the corresponding *NLR* clusters (*Seeholzer et al., 2010*; *Maekawa et al., 2019*; *Shen, 2003*). Isolate-specific disease resistance to *Bgh* mediated by MLA receptors is invariably linked to the activation of localized host cell death, and this immune response likely terminates growth of the biotrophic pathogen by shutting off its nutrient supply (*Boyd et al., 1995*). Of note, the *Mla* orthologs *Sr33* in wheat (*Periyannan et al., 2013*) and *Sr50* in rye (*Mago et al., 2015*) confer disease resistance to the stem rust pathogen *Puccinia graminis* f. sp. *tritici* (*Pgt*) isolate Ug99, a major threat to global wheat production. *Pgt* and *Bgh* belong to the Basidiomycota and Ascomycota phyla, respectively, indicating that MLA receptors can detect the presence of independently evolved avirulence effectors. Barley *Mla* and wheat *Pm3* both encode intracellular NLRs with an N-terminal coiled-coil (CC) domain but lack significant sequence relatedness (*Zhou et al., 2001*), whereas 23 allelic barley MLA resistance proteins exhibit >91% amino acid (aa) sequence identity (*Seeholzer et al., 2010*), and 17 deduced allelic wheat Pm3 receptors share >97% aa sequence identity (*Bhullar et al., 2010*). Diversifying selection among resistance alleles of *Mla* and *Pm3* is largely confined to regions encoding the C-terminal leucine-rich repeats (LRR) of the receptors (*Seeholzer et al., 2010*; *Maekawa et al., 2019*; *Bhullar et al., 2010*). The polymorphic MLA LRR is critical for effector detection as shown by a series of reciprocal domain swaps between MLA1 and MLA6 (*Shen, 2003*). Barley *Mla1* confers race-specific disease resistance to a *Bgh* isolate carrying the cognate avirulence gene $AVR_{a1}$ in transgenic *Arabidopsis thaliana*, suggesting ∼ 150 million years of evolutionary conservation of the underlying immune mechanism and potentially pointing to a direct recognition mechanism of $AVR_A$ effectors that is highly conserved between barley and *A. thaliana* (*Maekawa et al., 2012*).

Recently, the *Bgt* avirulence gene $AvrPm3^{a2/f2}$, recognized by the wheat *Pm3a* and *Pm3f* alleles, was identified by a map-based cloning approach and found to encode a typical CSEP that belongs to an effector family of 24 members (*Bourras et al., 2015*). The clonal nature of the haploid *Bgh* pathogen facilitated the identification of isolate-specific sequence variation in the transcriptomes of a global collection of 17 *Bgh* strains. These were associated with *Bgh* pathotypes on *Mla1*- and *Mla13*-containing near-isogenic barley lines to identify sequence-unrelated $AVR_{a1}$- and $AVR_{a13}$-encoding CSEPs (*Lu et al., 2016*). However, it remains unclear whether these powdery mildew avirulence effectors bind directly to the corresponding NLR receptors or whether the receptors sense the presence of the pathogen indirectly through effector-mediated modifications of host proteins.

To reveal the molecular mechanism underlying the co-evolutionary diversification of barley MLA NLRs with host-adapted *Bgh*, we first described the natural $AVR_a$ gene diversity in a local *Bgh* population. This allowed us to isolate four additional $AVR_A$ effectors by applying a transcriptome-wide association study (TWAS) with high-confidence single nucleotide polymorphisms (SNPs) underlying disease resistance/susceptibility phenotypes in barley. We then employed co-transfection experiments to show that only matching $Mla$-$AVR_a$ pairs activate cell death in barley leaf protoplasts. *Agrobacterium tumefaciens*-mediated co-delivery of $Mla$-$AVR_a$ pairs in *Nicotiana benthamiana* leaves demonstrated that their co-expression is sufficient to trigger cell death in this heterologous plant species. The $AVR_a$ avirulence effectors encode CSEPs that are sequence-unrelated except for allelic $AVR_{A10}$ and $AVR_{A22}$. These findings are inconsistent with a co-evolutionary arms race model

describing iterative cycles of MLA receptor and $AVR_A$ effector adaptations. Moreover, we failed to detect cell death activity for the previously reported $EKA\_AVR_{a10}$, a member of the EKA gene family that is derived from part of a class-1 LINE retrotransposon (*Ridout, 2006*; *Amselem et al., 2015*). Using co-expression experiments with matching $Mla$-$AVR_a$ pairs, including previously reported $AVR_{a13}$, we present evidence for direct receptor-avirulence effector interactions. Our findings imply that MLA receptors have an exceptional propensity to directly detect sequence-unrelated and likely structurally diverse pathogen effectors and that this feature might have facilitated the functional diversification of the receptor in the host population.

## Results

### Natural variation of $AVR_a$ genes within a local *Bgh* population

In 2016 we collected 13 *Bgh* field isolates from leaves of barley plants within the same locality near Cologne, Germany (GPS 5°57′N, 6°51′E 5; isolates designated K2, K3, K4, S11, S15, S16, S19, S20, S21, S22, S23, S25 and S26). We purified the isolates by serial propagation of conidiospores collected from single powdery mildew colonies (Materials and methods). Then, we determined their pathotypes (virulence/avirulence profiles) together with a global collection of 17 previously described *Bgh* isolates (*Lu et al., 2016*) on a panel of near-isogenic barley lines (NILs) in the cultivar (cv.) Pallas genetic background (*Kolster et al., 1986*) carrying *Mla1* (P01), *Mla3* (P02), *Mla6* (P03), *Mla7* (P04B), *Mla9* (P08B), *Mla10* (P09), *Mla12* (P10), *Mla22* (P12), or *Mla23* (P13) recognition specificities (*Supplementary file 1*). To assess infection types (ITs) independently, we additionally employed a panel of NILs in the genetic background of cv. Manchuria (*Moseman, 1972*) carrying introgressed *Mla1* (CI16137), *Mla6* (CI16151), *Mla7* (CI16147 and CI16153), *Mla10* (CI16149), or *Mla13* (CI16155) (*Supplementary file 1*). Similar to previous studies (*Seeholzer et al., 2010*; *Lu et al., 2016*), we scored five macroscopically distinguishable ITs at 9 d after conidiospore inoculation on leaves, ranging from 1 to 3 (=avirulent) or 4 to 5 (=virulent). Unambiguous assignment of virulent and avirulent interactions was possible for all isolates on *Mla1*, *Mla6*, *Mla9*, *Mla12*, *Mla13*, and *Mla22* NILs that displayed only IT 1, 2 or 5. A subset of the *Bgh* strains displayed IT 3 (avirulent) or 4 (virulent) on *Mla3*, *Mla7*, *Mla10*, and *Mla23* NILs, which were in some cases difficult to distinguish from each other and thus potentially complicating the discrimination of avirulent and virulent interactions with our experimental setup (*Supplementary file 1*). For two of the tested *Bgh* isolates different avirulent/virulent interactions were recorded between Pallas and Manchuria NILs carrying *Mla7* (isolates K4 and S22) and, similarly, two other isolates exhibited different avirulent/virulent interactions on Pallas and Manchuria NILs carrying *Mla10* (isolates CC88 and S20; *Supplementary file 1*). These few discrepancies of avirulent/virulent interactions might be explained by differences in the size of introgressed chromosomal segments from the respective donor lines in Pallas and Manchuria NILs that might contain other *R* genes besides *Mla*. Therefore, we omitted the IT data of isolates K4 and S22 on *Mla7* NILs and of CC88 and S20 isolates on *Mla10* NILs in the TWAS (see below) to identify $AVR_{a7}$ and $AVR_{a10}$ candidates, respectively. Among the 13 newly described *Bgh* isolates collected from the same geographic site, isolates S19 and S21, as well as S23, S25, and S26 exhibited identical avirulence/virulence patterns on the NIL test panel (*Supplementary file 1*). Thus, 10 out of 13 isolates showed unique avirulece/virulence patterns and each isolate carries at least five different $AVR_a$ genes (*Supplementary file 1*). These findings point to a high level of genetic variation among $AVR_a$ genes within the local *Bgh* population.

We obtained deep fungal transcriptomes (RNA-Seq) for the ten Cologne-derived field isolates with unique avirulent/virulent patterns during pathogenesis on susceptible barley leaves at 16 and 48 hr post conidiospore inoculation, resulting in 0.8 to 6 million sequenced and mapped *Bgh* read pairs (fragments) per sample (Materials and methods). RNA-seq reads of these isolates, combined with RNA-seq reads of the previously characterized global collection of *Bgh* isolates (*Lu et al., 2016*) and of reference isolate DH14, were aligned against the recently assembled near chromosome-level DH14 genome supplemented with updated gene models (*Frantzeskakis et al., 2018*). Subsequently, a collective set of sequence polymorphisms for all isolates was identified from the combined alignment data. To assess the potential influence of population structure among the isolates prior to TWAS, we performed a population structure analysis, for which we extracted a set of high-quality diallelic and synonymous single-nucleotide polymorphisms (SNPs) from the complete set of

polymorphisms (Methods). A principal component analysis (PCA) plot and a neighbour-joining tree including all isolates confirmed the Japanese isolate RACE1 to be exceptionally divergent and assigned the three Australian isolates to a distinct clade, whereas the ten new field isolates clustered mostly together with other European, Japanese, and USA isolates (*Figure 1A*, *Figure 1—figure supplement 1*). Although all newly described isolates were collected from a single geographic location, three of these (K2, K3, and S16) appear to form a subgroup that is distinct from the other European isolates (*Figure 1A*, *Figure 1—figure supplement 1*). Interestingly, the avirulence profiles of isolates K3 and S16 on the 11 tested *Mla* barley recognition specificities belong to distinct pathotype clusters (*Figure 1B*), despite an exceptionally high similarity of K3 and S16 at the transcriptome level (*Figure 1A*).

## Combined TWAS and *Bgh* genome analysis identified candidates for *AVR_{a7}, AVR_{a9}, AVR_{a10}, and AVR_{a22}*

Our TWAS analysis (*Supplementary file 2 - 4*) allowed us to identify several $AVR_a$ candidates. Manual re-inspection of these candidates (*Lu et al., 2016*), revealed that only transcript polymorphisms of the top candidates $AVR_{a7}$, $AVR_{a9}$, $AVR_{a10}$, and $AVR_{a22}$ exhibit tight linkage to the avirulence phenotypes of our *Bgh* strain panel on *Mla7*, *Mla9*, *Mla10*, and *Mla22* NILs, respectively. Two genes, *BLGH_06689* and *BLGH_06672*, encoding two identical copies of CSEP0059, were identified as top-ranking candidates for $AVR_{a7}$ (p=1.78E-05, *Supplementary file 4*). This is consistent with previous reports in which two $AVR_{a7}$ loci were inferred based on segregation analysis among the progeny of crosses between *Bgh* strains (*Skamnioti et al., 2008*; *Pedersen et al., 2002*). In the near chromosome-level genome assembly of isolate DH14 (*Frantzeskakis et al., 2018*) the two *CSEP0059* copies

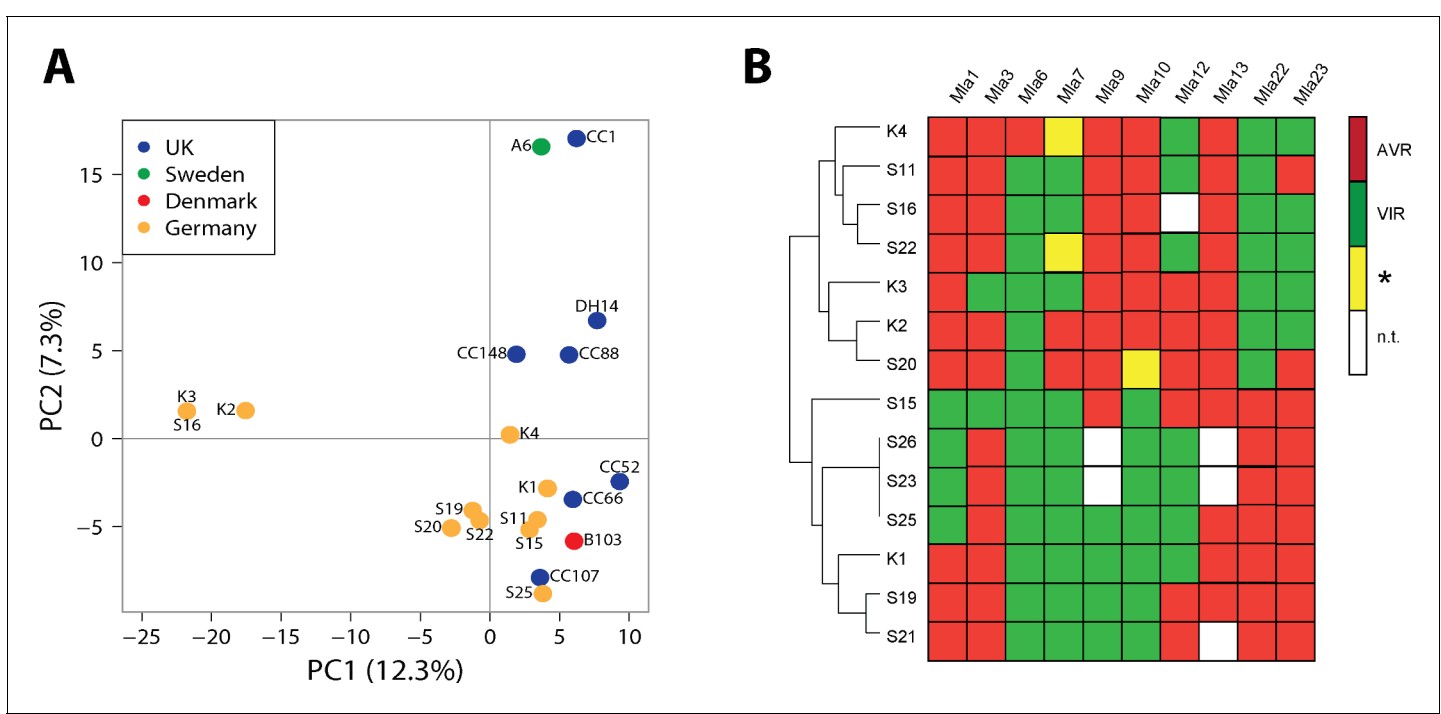

**Figure 1.** Population structure and avirulence profiles of *Blumeria graminis* f. sp. *hordei* (*Bgh*) isolates. (**A**) Principal Component Analysis (PCA) of the indicated European *Bgh* isolates, including ten newly collected strains from a local pathogen population in Germany, based on a set of 5170 diallelic high-quality synonymous SNPs. (**B**) Hierarchical clustering (R package 'pheatmap') of avirulence profiles from 14 *Bgh* isolates collected within the same area near Cologne, Germany (GPS 5°57′N, 6°51′E 5). Numbers correspond to infection types (ITs) 1, 2 and 3 = avirulent, red; infection types 4 and 5 = virulent, green. *denotes differences of ITs between cultivars Pallas and Manchuria, yellow. n.t.: not tested, white.

DOI: https://doi.org/10.7554/eLife.44471.003

The following figure supplement is available for figure 1:

**Figure supplement 1.** Population structure of a worldwide collection of 27 *Blumeria graminis* f. sp. *hordei* (*Bgh*) isolates.
DOI: https://doi.org/10.7554/eLife.44471.004

are physically separated by 141 kb on scaffold 44 and, together with four other *CSEPs*, form a *CSEP* cluster in this region (*Figure 2*). We designated the two identical *CSEP0059* copies in DH14 as candidate gene *AVR$_{a7}$-1* (*Figure 2A*). One of the DH14 *CSEPs* in the cluster is a *CSEP0059* paralog, *CSEP060* (*BLGH_06671*), which at the protein level shares 77% aa identity with CSEP0059 (*Figure 2B and C*). All virulent strains on *Mla7* NILs carry a SNP in *CSEP0059*, which results in a L51P substitution in the deduced candidate effector protein, and the corresponding variant was thus named AVR$_{a7}$-V1 (*Figure 2A and B*). Interestingly, only about half of the transcripts from the virulent isolate B103 *harbor* a SNP leading to the L51P substitution in the deduced protein. The other half of the B103 transcripts *harbor* a SNP resulting in a L28F substitution in the deduced protein (designated AVR$_{a7}$-V2), suggesting that in this strain the two *CSEP0059* copies differ. Similarly, approximately half of the transcript reads of all three avirulent Australian isolates (Art, Aby, and Will) harbor a SNP leading to an I65T substitution in the deduced protein (designated AVR$_{a7}$-AUS; *Figure 2A and B*), suggesting that these isolates also contain an even number of non-identical *CSEP0059* copies. Finally, the most divergent *Bgh* isolate RACE1 is clearly avirulent on *Mla7* NILs (*Supplementary file 1*), and in its transcriptome we found two non-synonymous SNPs in all *CSEP0059* transcripts corresponding to S47R and A96V aa substitutions in the deduced protein (designated AVR$_{a7}$-2; *Figure 2A and B*). Inspection of the near chromosome-level RACE1 genome (*Frantzeskakis et al., 2018*) revealed two identical *CSEP0059* gene copies (*BGHR1_17217 and BGHR1_17236*) separated by 151 kb on a single contig within a *CSEP* cluster that is largely syntenic with the corresponding DH14 *CSEP* cluster (*Figure 2C*; *Frantzeskakis et al., 2018*). RACE1 *CSEP0060* (*BGHR1_16533*) is located on the neighbouring contig tig00005324 and is possibly syntenic to the CSEP0060 location in DH14. Additionally, RACE1 harbors a third *CSEP0059* copy (*BGHR1_17237*) with S5N, M9T, T11A, L12W, L17F, S47R, L51P, N62S, A96V, and R105Q

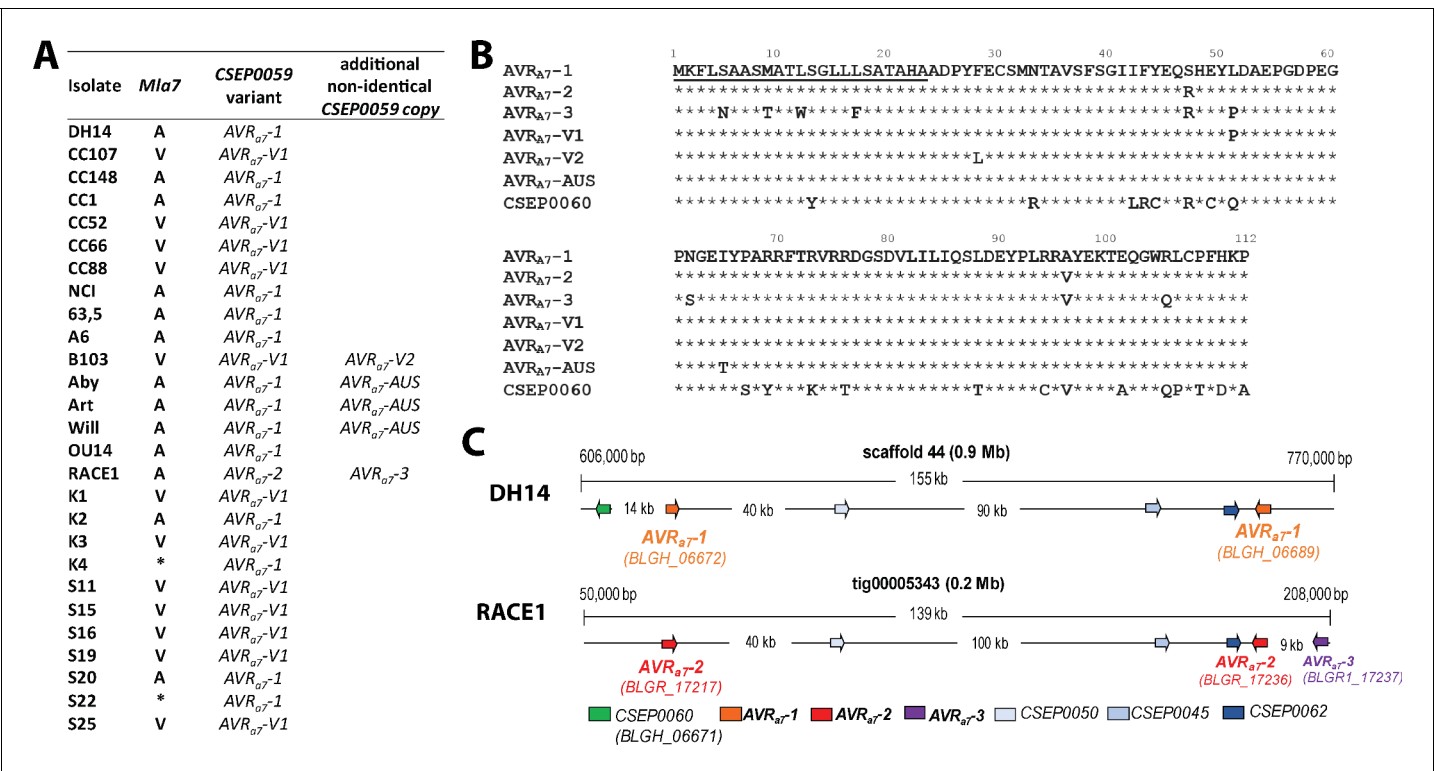

**Figure 2.** Identification of *CSEP0059* as an *AVR$_{a7}$* candidate by association of avirulence profiles with transcript polymorphisms and integration in the physical *Bgh* map. (A) *AVR$_{a7}$* transcript variants encoded by the indicated *Bgh* isolates with corresponding avirulence profiles. (B) Alignment of deduced AVR$_{A7}$ amino acid sequences with all variants highlighted. (C) Visualization of the chromosomal regions harboring *CSEP0059/AVR$_{a7}$* candidate variants with corresponding gene IDs in the genomes of *Bgh* isolates DH14 and RACE1. All *CSEPs* are depicted by arrows. * denotes different infection types between cultivars Pallas and Manchuria.

DOI: https://doi.org/10.7554/eLife.44471.005

substitutions in the deduced effector polypeptide (designated AVR$_{A7}$-3) (*Figure 2A, B and C*). Taken together, TWAS and the comparative genome analysis revealed the potential existence of multiple AVR$_{A7}$ variants, partly encoded by paralogous genes, and two virulent AVR$_{A7}$ variants.

TWAS identified *CSEP0174* (*BLGH_04994* in DH14 and *BGHR1_10042* in RACE1) as the top-ranking *AVR$_{a9}$* candidate (p=1.84E-05; *Supplementary file 4*, *Figure 3A*). Analysis of the transcriptome reads showed that isolates virulent on *Mla9* carry *AVR$_{a9}$* variants with either H28D (AVR$_{A9}$-V1) or G22E (AVR$_{A9}$-V2) substitutions (*Figure 3A and B*). Furthermore, we identified *CSEP0141* as the top-ranking candidate for both *AVR$_{a10}$* (*BGHR1_10013*, p=4.50E-07) and *AVR$_{a22}$* (p=1.22E-05) (*Supplementary file 4*, *Figure 3C*). All *Bgh* isolates avirulent on *Mla10* plants encode a CSEP0141 variant with an I77F substitution (AVR$_{A10}$) compared to the deduced effector protein in reference isolate DH14 (*BLGH_05021*; *Figure 3C and D*). By contrast, all isolates avirulent on *Mla22* plants carry a CSEP0141 variant with 13 deduced aa substitutions (V11A, F13L, D45G, D53E, Q55H, D58N, G59D, Q61P, H64Y, I77Y, V93L, W96L, and I111N; AVR$_{A22}$ in *Figure 3C and D*). DH14 is the only tested isolate that is virulent on both *Mla10*- and *Mla22*-expressing plants, and we therefore named the CSEP0141 variant encoded by DH14 *AVR$_{a10}$-V/AVR$_{a22}$-V*. CSEP0141 belongs to the CSEP family 64 with CSEP0266 being the only other family member that shares significant similarity with candidates AVR$_{A10}$ and AVR$_{A22}$ (59% and 63% identical aa sequences, respectively). Collectively, these data raised the possibility that MLA10 and MLA22 each recognize one of the two major natural variants of CSEP0141 in the *Bgh* population.

*AVR$_{a9}$* and *AVR$_{a10}$/AVR$_{a22}$* candidates are each single-copy genes in both DH14 and RACE1 *Bgh* genomes (*Frantzeskakis et al., 2018*). Notably, candidates *AVR$_{a10}$-V/AVR$_{a22}$-V* (*BLGH_05021*) in DH14 and *AVR$_{a10}$* (*BGHR1_10013*) in RACE1 physically locate approximately 300 kb away from the 3′ end of *AVR$_{a9}$* (*BLGH_04994* and *BGHR1_10042*) on the respective contiguous chromosomal DNA stretches (*Figure 3E*). The physical linkage of these candidate avirulence genes is consistent with a tight genetic coupling of the *AVR$_{a10}$* and *AVR$_{a22}$* loci and their linkage to the *AVR$_{a9}$* locus (*Skamnioti et al., 2008*). In addition, we found the transcript of candidate *AVR$_{a22}$* to map to a physical contig in the *Bgh* K1 genome assembly (tig19704; *Hacquard et al., 2013*) that is syntenic with the genomic region encompassing candidate *AVR$_{a10}$-V/AVR$_{a22}$-V* in DH14 and candidate *AVR$_{a10}$* in RACE1 (*Figure 3E*). The findings lend further support to the notion that *AVR$_{a10}$* and *AVR$_{a22}$* represent two alleles of a single *Bgh* effector gene (*Figure 3—figure supplement 1*). Candidate *AVR$_{a10}$*, which encodes a CSEP0141 variant, maps only 24 kb away from the 3′ end of the only other family member *CSEP0226* (*BLGH_05020* and *BGHR1_10014* in DH14 and RACE1 genomes, respectively) *and* approximately 60 kb away from the 5′ end of *one* of the EKA class-1 LINE retrotransposon family members (*Figure 3E*, *Amselem et al., 2015*). Reminiscent of the *AVR$_{a7}$*-containing cluster of sequence-unrelated effector genes in the *Bgh* genome (*Figure 2c*), the genomic region encompassing *CSEP0174* (*AVR$_{a9}$*) and *CSEP0141* (*AVR$_{a10}$/AVR$_{a22}$*) contains sequence-unrelated *CSEP0077*, *CSEP0080*, *CSEP0082*, *CSEP0085*, *CSEP0097*, and *CSEP0266*, and therefore represents another cluster of candidate secreted effector genes in the fungal genome.

## Functional analysis of *AVR$_a$* candidates in barley leaf protoplasts

We examined the capability of the identified AVR$_A$ candidates to trigger MLA-dependent cell death upon transient co-expression with matching MLA receptors in barley leaf protoplasts. All gene constructs were expressed from the *Zea mays* ubiquitin promotor together with the firefly luciferase (LUC) reporter gene (*Lu et al., 2016*), and relative LUC activity was quantified as a proxy for protoplast viability. As epitope tag sequences can interfere with signal-noise ratios of LUC activity in this assay (*Lu et al., 2016*), we refrained from fusion of constructs with epitope sequences.

When compared to the empty vector (EV) control, co-expression of previously reported *AVR$_{a1}$* (*Lu et al., 2016*) with *Mla1* reduced LUC activity in a *Mla1*-dependent manner by 71% (*Figure 4A*) *and co-expression of AVR$_{a7}$-1 with Mla7 significantly reduced the reporter activity by 37% but not when co-expressed with Mla1 (Figure 4A)*. Interestingly, co-expression of *Mla7* with either of the two *AVR$_{a7}$* variants present in RACE1, *AVR$_{a7}$-2* and *AVR$_{a7}$-3, resulted* in a 91% and 79% reduction of LUC activity, respectively, whereas their co-expression with *Mla1* did not significantly alter reporter activity (*Figure 4A*, *Figure 4—figure supplement 1A*). Taken together these data indicate that naturally occurring *AVR$_{a7}$* variants in the Bgh population differ in their propensity to activate MLA7-mediated cell death. Neither expression *of AVR$_{a7}$-V1 (CSEP0059) nor CSEP0060*, the paralog located adjacent to *AVR$_{a7}$-1* in the DH14 genome (*Figure 3C*), resulted in a significant *Mla7*-



**Figure 3.** Identification of *CSEP0174* as an *AVR~a9~* candidate and *CSEP0141* as an *AVR~a10~* and *AVR~a22~* candidate by association of avirulence profiles with transcript polymorphisms as well as integration in the physical *Bgh* map. (**A**) *AVR~a9~* variants encoded by each *Bgh* isolate with corresponding avirulence profiles. (**B**) Alignment of deduced AVR~A9~ amino acid sequences with variants highlighted. (**C**) *AVR~a10~/AVR~a22~* variants encoded by each *Bgh* isolate with corresponding avirulence profiles. (**D**) Alignment of AVR~A10~, AVR~A22~, and AVR~A10~-V/AVR~A22~-V amino acid sequences. (**E**) Visualization of the chromosomal regions harboring *CSEP0174/AVR~a9~* and *CSEP0141/AVR~a10~/AVR~a22~* candidates with corresponding gene IDs as well as a copy of the EKA family class-1 retrotransposon and other *CSEPs* in the genomes of *Bgh* isolates DH14, RACE1, and K1. *denotes differences of infection types between cultivars Pallas and Manchuria. n.t.: not tested.

DOI: https://doi.org/10.7554/eLife.44471.006

The following figure supplement is available for figure 3:

**Figure supplement 1.** Maximum likelihood phylogeny tree for the 805 predicted secreted proteins of *Bgh* DH14 lacking respective signal peptides.

DOI: https://doi.org/10.7554/eLife.44471.007

dependent reduction in reporter activity (*Figure 4A*, *Figure 4—figure supplement 1B*). We detected a 30% reduction in LUC activity when *AVR~a7~-AUS* was co-expressed with *Mla7* but not when co-expressed with *Mla1*. Co-expression of *AVR~a7~-V2* with *Mla7* did not result in significantly reduced LUC activity when compared to co-expression with *Mla1* (*Figure 4—figure supplement*

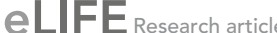

**Figure 4.** AVR$_A$ candidates induce MLA-specified cell death in transient gene expression assays. (A-C) Barley protoplasts were transfected with *pUBQ: luciferase* and either an piPKb002 empty vector control (EV) or piPKb002 containing cDNAs of (A) *AVR$_{a1}$, AVR$_{a7}$* variants and *CSEP0060,* all lacking their respective signal peptides (SPs) together with either *Mla1* or *Mla7*; (B) *AVR$_{a9}$* variants and *AVR$_{a13}$* variants, all lacking their respective SPs together with either *Mla9* or *Mla13*; (C) *AVR$_{a10}$, AVR$_{a10}$-V/AVR$_{a22}$-V, AVR$_{a22}$,* and *CSEP0266* without SPs and *EKA_AVR$_{a10}$*, together with either *Mla10* or *Mla22*. Luciferase activity was determined ~16 hr post transfection as a proxy for cell death and normalized for each *Mla* construct by setting the detected luminescence for the corresponding EV transfection to 1. All values obtained in at least four independent experiments are indicated by dots, error bars = standard deviation. Differences between samples were assessed by analysis of variance (ANOVA) and subsequent Tukey post hoc tests for each *Mla* construct. Calculated *P* values were as follows: *Mla7*: $p=1.2e\text{-}13$ and *Mla1*: $p=1.9e\text{-}03$ (A); *Mla13*: $p=3.2e\text{-}06$ and *Mla9*: $p=1.3e\text{-}04$ (B); *Mla10*: $p=4.2e\text{-}04$ and *Mla22*: $p=5.5e\text{-}04$ (C). Samples marked by identical letters in the plots do not differ significantly ($p<0.05$) in the Tukey test for the corresponding transfected *Mla*. (D–H) *Nicotiana benthamiana* plants were transformed transiently as indicated. cDNAs lacking stop codons were fused in between the *35S* promotor sequence and *4xMyc* (*Mla* variants) or *mYFP* (*CSEPs* and *AVR$_a$* variants lacking SPs and *EKA_AVR$_{a10}$*) epitope sequences. Cell death (D, G) was determined three days post transformation and figures shown are representatives of at least three independently performed experiments with at least three transformations per experiment. Protein levels (E, F, H) of MLA-4xMyc, CSEP-mYFP, AVR$_A$-mYFP and EKA_AVR$_A$-mYFP in *Nicotiana benthamiana* corresponding to constructs of D and G. Leaf tissue was harvested two days post infiltration. Total protein was extracted and recovered

*Figure 4 continued on next page*

*Figure 4 continued*

by GFP-Trap pull-down as indicated. Extracts and immunoprecipitates were separated by gel electrophoresis and probed by anti-Myc or anti-GFP western blotting (WB) as indicated. IP: Immunoprecipitated fraction. CBB: Coomassie brilliant blue.

DOI: https://doi.org/10.7554/eLife.44471.008

The following source data and figure supplements are available for figure 4:

**Source data 1.** Data points indicating relative luciferase activity of *Figure 4A*.

DOI: https://doi.org/10.7554/eLife.44471.011

**Source data 2.** Data points indicating relative luciferase activity of *Figure 4B*.

DOI: https://doi.org/10.7554/eLife.44471.012

**Source data 3.** Data points indicating relative luciferase activity of *Figure 4C*.

DOI: https://doi.org/10.7554/eLife.44471.013

**Figure supplement 1.** MLA-mediated cell death in barley protoplasts.

DOI: https://doi.org/10.7554/eLife.44471.009

**Figure supplement 1—source data 1.** Data points indicating relative luciferase activity of *Figure 4—figure supplement 1A*.

DOI: https://doi.org/10.7554/eLife.44471.014

**Figure supplement 1—source data 2.** Data points indicating relative luciferase activity of *Figure 4—figure supplement 1C*.

DOI: https://doi.org/10.7554/eLife.44471.015

**Figure supplement 1—source data 3.** Data points indicating relative luciferase activity of *Figure 4—figure supplement 1D*.

DOI: https://doi.org/10.7554/eLife.44471.016

**Figure supplement 2.** $AVR_A$-mYFP candidates induce MLA-4xMyc mediated cell death.

DOI: https://doi.org/10.7554/eLife.44471.010

**Figure supplement 2—source data 4.** Ceall death scores of samples shown in *Figure 4D* according to scoring system of *Figure 4—figure supplement 2B*.

DOI: https://doi.org/10.7554/eLife.44471.017

**Figure supplement 2—source data 5.** Ceall death scores of samples shown in *Figure 4G* according to scoring system of *Figure 4—figure supplement 2B*.

DOI: https://doi.org/10.7554/eLife.44471.018

*1A*). Barley accessions expressing the MLA7 polypeptide sequences CI1647 and CI1653 from Manchuria MLA7 NILs were found to differ slightly in two independent earlier studies (MLA7_AAQ55540, *Halterman and Wise, 2004*; MLA7 *Seeholzer et al., 2010*; *Figure 4—figure supplement 1B*). Therefore, we also tested the capability of MLA7_AAQ55540 to recognize $AVR_{a7}$ and found that co-expression of *MLA7_AAQ55540* with $AVR_{a7}$-2 reduced LUC activity by only 68% (as compared to 91% when co-expressed with *Mla7*). Luciferase activity in protoplasts co-expressing *MLA7_AAQ55540* and $AVR_{a7}$-1 (30% LUC reduction compared to EV) did not differ significantly from protoplasts co-expressing *MLA7_AAQ55540* and $AVR_{a7}$-V1 (15% LUC reduction compared to EV; *Figure 4—figure supplement 1C*).

Co-expression of *Mla9* with the $AVR_{a9}$ candidate, but not its variants $AVR_{a9}$-V1 and $AVR_{a9}$-V2 or $AVR_{a13}$-1 and $AVR_{a13}$-V2 (*Lu et al., 2016*), resulted in a significant 67% relative reduction of LUC activity. This cell death activity was specific to *Mla9* as co-expression of $AVR_{a9}$ *with Mla13* did not significantly reduce LUC activity (*Figure 4B*). Co-expression of the previously reported $AVR_{a13}$ (*Lu et al., 2016*) with *Mla13* reduced LUC activity in a *Mla13*-dependent manner by 69% (*Figure 4B*).

Similar to the moderate reduction of LUC activity when $AVR_{a7}$-1 was co-expressed with *Mla7* (*Figure 4A*), co-expression of the $AVR_{a10}$ candidate with *Mla10* reduced the reporter activity by only 25% (*Figure 4C*). LUC reduction was not observed when $AVR_{a10}$-V/$AVR_{a22}$-V, $AVR_{a22}$, CSEP0266, or EKA_$AVR_{a10}$ were co-expressed with *Mla10*. Co-expression of EKA_$AVR_{a10}$, $AVR_{a10}$, and *Mla10* had no impact on LUC reduction when compared to co-expression of $AVR_{a10}$ and *Mla10* alone (*Figure 4—figure supplement 2D*). Co-expression of *Mla22* with $AVR_{a22}$, but not $AVR_{a10}$, $AVR_{a10}$-V/ $AVR_{a22}$-V, CSEP0266, or EKA_$AVR_{a10}$, reduced relative LUC activity by 53% when compared to co-expression of *Mla22* with an EV control (*Figure 4C*). These findings provide functional evidence that *CSEP0141* alleles define $AVR_{a10}$ and $AVR_{a22}$ avirulence effectors and are specifically recognized by allelic MLA10 and MLA22 receptors, respectively.

## Co-expression of matching *Mla* and *AVR$_a$* pairs is necessary and sufficient to trigger cell death in *N. benthamiana*

Next we tested whether *Agrobacterium tumefaciens*-mediated delivery and co-expression of matching *Mla* and *AVR$_a$* pairs can trigger cell death in heterologous *N. benthamiana* leaves. In addition to newly isolated AVR$_{A7}$, AVR$_{A9}$ and AVR$_{A10}$/AVR$_{A22}$, we included the previously reported AVR$_{A1}$ and AVR$_{A13}$ variants as additional specificity controls in these experiments (*Figure 4D–4H*) (*Lu et al., 2016*). For protein expression and stability analysis in this heterologous system, the constructs were designed to express C-terminally 4xMyc-tagged MLA receptors and C-terminally mYFP-tagged AVR$_A$ variants without signal peptide sequences.

Delivery of the *AVR$_{a1}$-mYFP* construct but not the construct for its virulent variant, *AVR$_{a1}$-V1-mYFP*, conferred cell death in *N. benthamiana* when co-expressed with *Mla1-4xMyc*, but not when co-expressed with *Mla7-4xMyc* or *Mla13-4xMyc* (*Figure 4D*, *Figure 4—figure supplement 2*). We also observed statistically significant cell death intensity when *AVR$_{a13}$-1-mYFP* and *AVR$_{a13}$-3-mYFP*, but not the virulent variants *AVR$_{a13}$-V1-mYFP* or *AVR$_{a13}$-V2-mYFP* when co-expressed with *Mla13-4xMyc* (*Figure 4D*; *Figure 4—figure supplement 2*). Cell death was not seen when the same *AVR$_a$* effector constructs were co-expressed with *Mla1-4xMyc* or *Mla7-4xMyc* (*Figure 4D*; *Figure 4—figure supplement 2*), indicating retained recognition specificity of MLA1 and MLA13 receptors in this heterologous plant species, respectively.

*AVR$_{a7}$-1* mediates moderately reduced LUC reporter activity in barley protoplasts when co-expressed with *Mla7*, whereas *AVR$_{a7}$-2* expression leads to a strong reduction of reporter activity in the same experiment (*Figure 4A*). Correspondingly, in *N. benthamiana* we observed MLA7-dependent cell death only when expressing *AVR$_{a7}$-2-mYFP* together with *Mla7-4xMyc*, but not when *Mla7-4xMyc* was co-expressed with *AVR$_{a7}$-1-mYFP* or *AVR$_{a7}$-V1-mYFP* variants (*Figure 4D*, *Figure 4—figure supplement 2*). The lack of AVR$_{A7}$-1 - MLA7 cell death in *N. benthamiana* is not due to AVR$_{A7}$-1-mYFP protein stability (*Figure 4E*) but may be due to other unknown aspects of the heterologous system.

Major differences in protein steady-state levels were found between individual AVR$_A$ effectors, whereas protein levels of all MLA receptors were comparable in α-Myc western blots (*Figure 4E and F*). mYFP-tagged AVR$_{A1}$ variants were barely detectable even after enrichment by GFP-Trap (*Figure 4E*), whilst AVR$_{A13}$-1, AVR$_{A13}$-3, *and* AVR$_{A13}$-V2 were detectable in *N. benthamiana* extracts without GFP-Trap pull-down (*Figure 4E*). AVR$_{A13}$-V1-mYFP protein was barely detectable even after GFP-Trap enrichment (*Figure 4E*) suggesting that loss of MLA13-mediated cell death activity for AVR$_{A13}$-V1 may be due to protein instability. Notably, we found a Western blot signal corresponding to the expected size of ~40 kDa for the AVR$_{A13}$-V2-mYFP fusion protein (*Figure 4E*). This differs from the expression of *FLAG-AVR$_{a13}$-V2* in barley protoplasts, where most of the FLAG-AVR$_{A13}$-V2 fusion protein was visible as a cleaved protein product (*Lu et al., 2016*). mYFP-tagged AVR$_{A7}$ variants were only detectable in *N. benthamiana* leaf extracts after GFP-Trap pull-downs (*Figure 4E*).

*AVR$_{a9}$* elicited a 67% reduction in LUC activity when co-expressed with *Mla9* in barley protoplasts (*Figure 4B*). Surprisingly however, in *N. benthamiana*, neither AVR$_{A9}$-mYFP nor its virulent variants AVR$_{A9}$-V1-mYFP or AVR$_{A9}$-V2-mYFP triggered MLA9-dependent cell death (*Figure 4G*; *Figure 4—figure supplement 2*) although mYFP-tagged AVR$_{A9}$ was detectable in *N. benthamiana* protein extracts (*Figure 4H*). We postulate either that a third barley protein other than MLA9 and AVR$_{A9}$ is needed for MLA9-mediated cell death activation in *N. benthamiana* or that the chosen combination of epitope tags prevents effective AVR$_{A9}$ recognition in this plant species.

Although *AVR$_{a10}$* reduced LUC reporter activity in barley protoplasts only moderately when co-expressed with *Mla10* (*Figure 4C*), we detected a clearly visible cell death response when *AVR$_{a10}$-mYFP* was co-expressed with *Mla10-4xMyc* in *N. benthamiana* leaves in multiple independent experiments (*Figure 4G*; *Figure 4—figure supplement 2*). The MLA10-triggered cell death specifically required the presence of AVR$_{A10}$-mYFP because leaf cells remained alive upon co-expression with AVR$_{A10}$-V/AVR$_{A22}$-V-mYFP or AVR$_{A22}$-mYFP (*Figure 4G*; *Figure 4—figure supplement 2*). Consistent with data obtained with barley leaf protoplasts, cell death in heterologous *N. benthamiana* was induced upon co-expression of *Mla22-4xMyc* with *AVR$_{a22}$-mYFP*, but not *AVR$_{a10}$-mYFP* or *AVR$_{a10}$-V/AVR$_{a22}$-V-mYFP* (*Figure 4G*; *Figure 4—figure supplement 2*). Furthermore, transient expression of *CSEP0266-mYFP* or *EKA_AVR$_{a10}$-mYFP* together with either *Mla10-4xMyc* or *Mla22-4xMyc* failed to trigger cell death in *N. benthamiana* (*Figure 4G*; *Figure 4—figure supplement 2*), corroborating

our conclusion that allelic AVR$_{A10}$ and AVR$_{A22}$ avirulence effectors are specifically recognized by allelic MLA10 and MLA22 receptors, respectively. Only AVR$_{A10}$-mYFP and AVR$_{A10}$-V/AVR$_{A22}$-V-mYFP were detectable in *N. benthamiana* protein extracts and these two fusion proteins as well as CSEP0266-mYFP and AVR$_{A22}$-mYFP ran at the expected size of ~40 kDa after enrichment by GFP-Trap pull-downs (*Figure 4H*). In these experiments the AVR$_{A22}$-mYFP protein appears less stable than AVR$_{A10}$-mYFP (*Figure 4H*). EKA_AVR$_{A10}$-mYFP is expected to migrate at ~70 kDa. We detected a western blot signal at this expected size and one faster-migrating variant (*Figure 4H*). In conclusion, we failed to detect cell death activity in response to the previously reported recognition of *EKA_AVR$_{a10}$* by MLA10 (*Ridout, 2006*). This contrasts with the functional validation of the *AVR$_{a10}$* candidate identified here by TWAS.

## Candidate AVR$_A$ proteins interact with matching MLA receptors in plant extracts and in yeast

AVR$_{A1}$ is recognised by MLA1 in barley, *A. thaliana* and, as shown here, also in *N. benthamiana* (*Figure 4D*; *Lu et al., 2016*). The retention of MLA1-dependent recognition of AVR$_{A1}$ in three divergent plant families (Triticeae, Brassicaceae, Solanaceae) suggests direct interactions of matching MLA and AVR$_A$ pairs or an indirect recognition mechanism involving highly evolutionarily conserved AVR$_A$ host target(s). The wheat *Mla* ortholog Sr50 interacts with its cognate effector AvrSr50 of *Puccinia graminis* f. sp. *tritici* in yeast (*Chen et al., 2017*). Despite the lack of sequence conservation between most of the identified AVR$_A$ effectors and AvrSr50, we tested whether barley MLA directly interacts with cognate *Bgh* effectors. So far it has been impossible to purify large quantities of recombinant full-length MLA receptors for in vitro AVR$_A$-MLA association studies, possibly because of MLA-triggered cell death and receptor oligomerisation (*Maekawa et al., 2011b*). We thus focused on quantitatively measuring putative AVR$_A$-MLA associations in plant extracts using the highly sensitive split-luciferase (split-LUC) complementation assay (*Paulmurugan et al., 2002*; *Luker et al., 2004*). Whereas barley protoplasts can undergo cell death upon expression of matching *Mla* and *AVR$_a$* pairs at ~16 hr post transfection (*Figure 4A–4C*), expression in *N. benthamiana* leaves permitted MLA and AVR$_A$ interaction analysis at two days post *A. tumefaciens* leaf infiltration and prior to the appearance of macroscopically visible MLA-mediated cell death. To examine MLA and AVR$_A$ associations by luciferase activity of protein extracts from *A. tumefaciens*-infiltrated leaf area, we generated gene constructs in which MLA and AVR$_A$ were fused at the C-terminus to cLUC and nLUC, respectively (*Mla-cLUC* and *AVR$_a$-nLUC*), and used these for *A. tumefaciens*-mediated transient gene expression experiments in *N. benthamiana* leaves (Materials and methods). We focused on *AVR$_{a13}$*, *AVR$_{a7}$*, and *AVR$_{a10}$/AVR$_{a22}$* variants and their cognate *Mla* receptors for split-LUC assays (*Figure 5A–5C*), because AVR$_{A1}$ protein levels were barely detectable (*Figure 4E*), and co-expression of *Mla9* and *AVR$_{a9}$* failed to trigger a cell death response in *N. benthamiana* leaves (*Figure 4G*).

We detected high LUC activities (>10,000 units) in extracts of leaves expressing *Mla13-cLUC* together with *AVR$_{a13}$-1-nLUC* and *AVR$_{a13}$-3-nLUC* but not when *Mla13-cLUC* was exchanged to *Mla1-cLUC* (*Figure 5A*), although the two MLA-cLUC proteins were similarly stable (*Figure 5D*). This was not the case for samples expressing *AVR$_{a13}$-V1-nLUC*, possibly due to lack of detectable AVR$_{a13}$-V1-nLUC protein (*Figure 5E*). Unexpectedly, the highest LUC activity was seen when *Mla13-cLUC* was expressed together with *AVR$_{a13}$-V2-nLUC* (>100,000 units; *Figure 5A*). The high LUC activities were dependent on the MLA13 receptor because only low LUC activities (<300 units) were observed when the same *AVR$_{a13}$* variants were co-expressed with *Mla1-cLUC* (*Figure 5A*).

We detected LUC activity when co-expressing *Mla7-cLUC* with *AVR$_{a7}$-2-nLUC*, which was at least 7-fold higher when the latter construct was replaced by *AVR$_{a7}$-1-nLUC* or *AVR$_{a7}$-V1-nLUC* (1000 units; *Figure 5B*), despite comparable protein levels of all AVR$_{A7}$-nLUC variants (*Figure 5E*). The LUC activities were much lower compared to the LUC complementation of *Mla13-cLUC* and *AVR$_{a13}$-1-nLUC* (*Figure 5A*). The higher LUC reporter activity was not detected when co-expressing *Mla13-cLUC* with *AVR$_{a7}$-2-nLUC*. This proxy for *in planta* receptor-avirulence effector interaction is in agreement with our observation that AVR$_{A7}$-2-mYFP but not AVR$_{A7}$-1-mYFP or AVR$_{A7}$-V1-mYFP is capable of inducing MLA7-dependent cell death in *N. benthamiana* (*Figure 4D*).

Co-expression of *AVR$_{a10}$-nLUC* with *Mla10-cLUC* resulted in the detection of a 3.5-fold higher LUC activity when compared to expression of its virulent variants *AVR$_{a10}$-V/AVR$_{a22}$-V-nLUC* and *AVR$_{a22}$-nLUC* (*Figure 5C*). LUC activity upon expression of *AVR$_{a22}$-nLUC* together with *Mla22-cLUC*

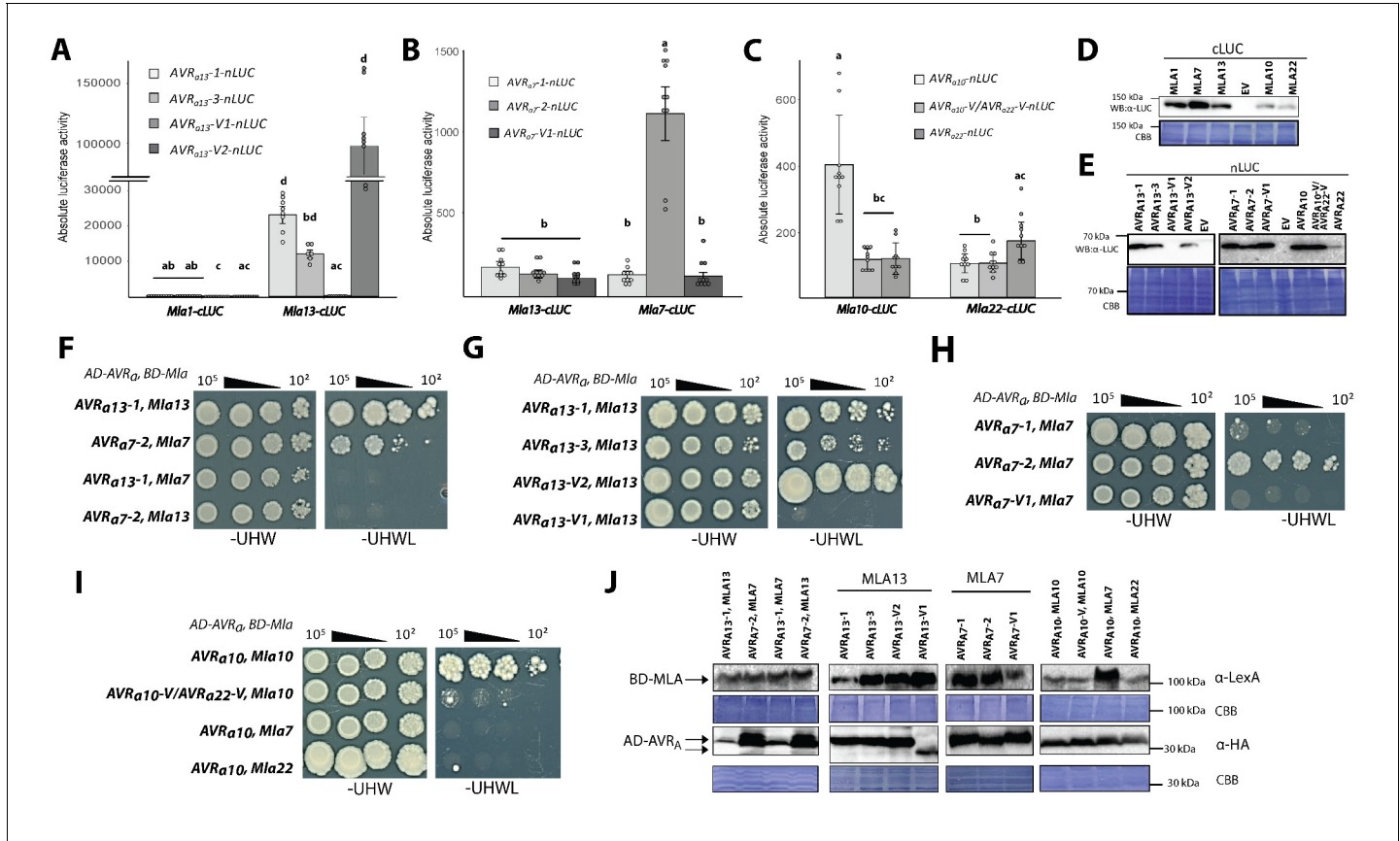

**Figure 5.** Association of candidate AVR$_A$ proteins with MLA in plant extracts (A–E) and in yeast (F–J). (A-C) *Nicotiana benthamiana* plants were transformed transiently with constructs encoding (**A**) *Mla1-cLUC* or *Mla13-cLUC* together with cDNAs of *AVR$_{a13}$-1*, *AVR$_{a13}$-3*, *AVR$_{a13}$-V1*, *AVR$_{a13}$-V2* lacking signal peptides (SPs) and fused C-terminally in frame with nLUC, (**B**) *Mla1-cLUC*, or *Mla7-cLUC* together with cDNAs of *AVR$_{a7}$-1*, *AVR$_{a7}$-2*, and *AVR$_{a7}$-V1* lacking SPs and fused C-terminally in frame with nLUC, (**C**) *Mla10-cLUC* or *Mla22-cLUC* together with cDNAs of *AVR$_{a10}$*, *AVR$_{a10}$-V/AVR$_{a22}$-V*, and *AVR$_{a22}$* without SPs fused C-terminally in frame with nLUC, all under the control of the 35S promotor. Luciferase activity was determined two days post transfection. All values obtained in at least six experiments are indicated by dots, error bars = standard deviation. For each graph, differences between samples were assessed using non-paramatric analysis of variance (Kruskal-Wallis) and subsequent Dunn's post hoc tests. Calculated *P* values were as follows: (**A**) p=6.8e-10, (**B**) p=1.2e-04, (**C**) p=8.0e-07. Samples marked by identical letters in the plot did not differ significantly (p<0.05) in Dunn's test. (**D–E**) Protein levels of MLA-cLUC (**D**) and AVR$_A$-nLUC (**E**) variants in *Nicotiana benthamiana* corresponding to constructs of 5A to 5C. Leaf tissue was harvested two days post infiltration. Total protein was extracted, separated by gel electrophoresis and probed by anti-LUC western blotting (WB). (**E–I**) Yeast cells were co-transformed with *Mla* alleles fused N-terminally to the LexA binding domain (BD) and *AVR$_a$* constructs lacking SPs fused N-terminally to the B42 activation domain (AD) and 1xHA tag as indicated. Growth of transformants was determined on selective growth media containing raffinose and galactose as carbon sources but lacking uracil, histidine and tryptophan (-UHW), and interaction of proteins was determined by leucine reporter activity reflected by growth of yeast on selective media containing raffinose and galactose as carbon sources but lacking uracil, histidine, tryptophan and leucine (-UHWL). Figures shown are representatives of at least three independent experiments with yeast clones obtained from three independent yeast transformation experiments and pictures were taken 12 days after drop out. (**J**) Protein levels of BD-MLA and AD-AVR$_A$ variants corresponding to yeast of *Figure 5D–5G*. Yeast transformants were grown in glucose containing selective media lacking uracil, tryptophan, and histidine to OD$_{600}$ = 1. Cells were harvested, total protein extracted, separated by gel electrophoresis, and western blots (WB) were probed with anti-LexA or anti-HA antibodies as indicated.

DOI: https://doi.org/10.7554/eLife.44471.019

The following source data is available for figure 5:

**Source data 1.** Data points indicating absolute luciferase activity of *Figure 5A*.
DOI: https://doi.org/10.7554/eLife.44471.020

**Source data 2.** Data points indicating absolute luciferase activity of *Figure 5B*.
DOI: https://doi.org/10.7554/eLife.44471.021

**Source data 3.** Data points indicating absolute luciferase activity of *Figure 5C*.
DOI: https://doi.org/10.7554/eLife.44471.022

was only slightly higher (1.6-fold) when compared to its virulent variants $AVR_{a10}$-nLUC and $AVR_{a10}$-V/ $AVR_{a22}$-V-nLUC in the same experiment (**Figure 5C**). This marginal increase in LUC activity may partly reflect differences in protein stability, as the $AVR_{A22}$-nLUC protein is barely detectable when compared to $AVR_{A10}$-nLUC (**Figure 5E**) and because both, MLA10-cLUC and MLA22-cLUC are seemingly less stable when compared to MLA7-cLUC and MLA13-cLUC.

Although the LUC complementation assay is suggestive of direct receptor – avirulence effector associations, we cannot fully exclude the involvement of other plant proteins in this association. We thus tested MLA and AVR$_A$ interactions in a yeast two-hybrid assay using leucine reporter gene activity (**Figure 5F–5I**). Similar to results obtained with the split-LUC assay, yeast growth on leucine-deficient medium was observed when we co-expressed *AD* (B42 activation domain)-$AVR_{a13}$-1 with *BD* (LexA binding domain)-*Mla13* and *AD*-$AVR_{a7}$-2 with *BD*-*Mla7*, but not when *BD*-*Mla13* and *BD*-*Mla7* were swapped in these two interaction experiments, indicating the interactions are specific for matching MLA receptor and avirulence effector pairs (**Figure 5F**). Interactions in yeast were detectable when *BD*-*Mla13* was co-expressed with either $AVR_{a13}$-1 or $AVR_{a13}$-3, but undetectable upon co-expression with $AVR_{a13}$-V1, suggesting specific MLA13 interactions with the avirulent AVR$_{A13}$ variants (**Figure 5G**). However, clear interactions in yeast were also detectable upon co-expression of *BD*-*Mla13* with *AD*-$AVR_{a13}$-V2, even at a cell plating density of $10^2$ (**Figure 5E**), which is reminiscent of the split-LUC result with the corresponding construct pair (**Figure 5A**). In contrast to AVR$_{A13}$-V1-mYFP (**Figure 4E**) and AVR$_{A13}$-V1-nLUC (**Figure 5E**) protein in *N. benthamiana* leaves, AD-AVR$_{A13}$-V1 protein level in yeast was comparable to other AD-AVR$_{A13}$ variants (**Figure 5J**). Robust interaction in yeast was found upon co-expression of *BD*-*Mla7* with *AD*-$AVR_{a7}$-V2, but not when the former was co-expressed with *AD*-$AVR_{a7}$-V1 (**Figure 5H**). Only sporadic yeast colony growth was found when *BD*-*Mla7* was co-expressed with *AD*-$AVR_{a7}$-1 (**Figure 5H**). This again mirrors the findings with the corresponding gene pairs in the LUC complementation assay (**Figure 5B**). We conclude that specific interactions can be detected for the matching MLA7 and AVR$_{A7}$-2 pair in yeast and specific associations for the corresponding protein pair in plant extracts. In sum, yeast two-hybrid and split-LUC experiments suggest direct detection of the sequence-unrelated avirulence effectors AVR$_{A7}$ and AVR$_{A13}$ by matching MLA7 and MLA13 receptors, but the strong association of the virulent effector AVR$_{A13}$-V2 with MLA13 represents one case in which receptor-effector association is uncoupled from receptor activation, that is from cell death induction (see Discussion).

Consistent with a direct interaction of matching MLA and AVR$_A$ pairs, co-expression of *BD*-*Mla10* with *AD*-$AVR_{a10}$ in yeast resulted in leucine reporter gene activation and this was undetectable when *Mla10* was replaced by either *Mla7* or *Mla22*, and minor when *AD*-$AVR_{a10}$ was replaced by its virulent variant *AD*-$AVR_{a10}$-V/$AVR_{a22}$-V (**Figure 5I**). The latter virulent effector differs only by a single amino acid from the avirulence effector AD-AVR$_{A10}$ (**Figure 3D**).

## Evolutionary history of *Bgh* AVR$_a$ genes and population-level AVR$_{a10}$/ AVRa$_{a22}$ sequence variation in *B. graminis formae speciales*

We used a high-quality genome assembly of the wheat powdery mildew *Bgt* reference isolate 96224 to investigate the evolutionary history of *Bgh* AVR$_a$ genes and to potentially identify distinctive selection pressures on these effector genes exerted by wheat and barley hosts. *AVR$_{a7}$* (sequence identical *BLGH_06672* and *BLGH_06689*; **Figure 2**) belongs to a larger *CSEP* gene family in the *Bgh* DH14 genome and, together with three closely related members, defines one sublineage (**Figure 6—figure supplement 1A**). Although we identified four genes sharing between 40% and 47% aa sequence similarity with *AVR$_{a7}$* (**Figure 6—figure supplement 1A**), the flanking genes of *AVR$_{a7}$* differ from those adjacent to the four *Bgt* homologs and therefore do not permit conclusions on potential orthologous relationships after the split of the two *formae speciales*.

We found three *Bgt* genes sharing between 52% and 57% identical polypeptide sequences with *Bgh* AVR$_{a9}$ (*BLGH_04994*) (**Figure 6—figure supplement 1C and D**). These genes are closely located to each other in a region of the wheat powdery mildew genome that is largely collinear between *Bgt* and *Bgh* genomes (**Figure 6—figure supplement 1E and F**). We conclude that either local gene duplications of an ancestral ortholog of *BLGH_04994* gave rise to the extant *Bgt* gene organization, or that these duplications were already present in the last common ancestor of *Bgt* and *Bgh* and that paralogs were lost in the *Bgh* genome.

Applying a phylogenetic approach to the wheat powdery mildew genome, we identified *BgtE-5921* as the ortholog of *Bgh* CSEP0141 (with haplotypes AVR$_{a10}$ and AVR$_{a22}$), which shares 68% and

65% identical deduced polypeptide sequences with AVR$_{A10}$ and AVR$_{A22}$, respectively (*Figure 6—figure supplement 1G*; *Table 1*; *Figure 6A*). In addition, we found a single *CSEP0141* candidate ortholog in each of the genomes of a world-wide collection of other *B. graminis formae speciales* (*Table 1*, *Supplementary file 5*), for which short-read genome sequences are available (*Spanu et al., 2010*; *Wicker et al., 2013*; *Hacquard et al., 2013*; *Menardo et al., 2016*; *Praz et al., 2017*; *Müller et al., 2019*). With these short-read genome sequences from multiple isolates of each of these *B. graminis* f. sp. (*Supplementary file 5*) we were able to assess genome-wide nucleotide diversity (π) and population-level sequence diversification of the *CSEP0141* orthologs and compared them with the diversification pattern of *CSEP0141* in the *Bgh* population. Based on 1141 neutral markers, we calculated a genome-wide per-site nucleotide diversity (π) of 0.022 for the f. sp. *hordei*, 0.013 for the f. sp. *secalis*, 0.050 for the f. sp. *tritici2*, 0.062 for the f. sp. *tritici*, and 0.040 for the f. sp. *triticale* isolates (*Supplementary file 5*). Unlike the two major *CSEP0141* haplotypes in *Bgh* (*AVR$_{a10}$* and *AVR$_{a22}$*), a single dominant *BgtE-5921* haplotype was found in the *Bgt* population (29 out of 40 isolates; *Table 1*, *Figure 6A*). The remaining 11 *Bgt* isolates represent six further *BgtE-5921* haplotypes encoding effector variants with at most four aa polymorphisms in the deduced proteins compared to the dominant BgtE-5921 variant (*Table 1*). Three of the latter haplotypes are exclusively present in *Bgt* isolates that were collected from tetraploid wheat and represent a distinctive *Bgt* sublineage, designated *B. graminis* f. sp. *tritici2* (*Menardo et al., 2016*). A single haplotype of the *CSEP0141* ortholog was detected among 22 *B. graminis* f. sp. *triticale* isolates and a single haplotype was also found in five *B. graminis* f. sp. *secalis* isolates (*Table 1*), indicating either limited or no natural variation of this effector in populations of wheat, triticale, and rye powdery mildews (up to four aa substitutions in wheat powdery mildews). This underlines the exceptional level of polymorphism between the two major *CSEP0141* haplotypes, *AVR$_{a10}$* and *AVR$_{a22}$*, in the *Bgh* population with 13 deduced aa changes. *Bgh CSEP0141* defines one of 190 core effector genes that are conserved across *B. graminis* ff. spp. (*Frantzeskakis et al., 2018*) and exhibits the highest frequency of non-synonymous SNPs in the *Bgh* population (4.2 non-synonymous SNPs/100 bp coding sequence; *Figure 6C*). Virulence functions of core effectors are likely important for *B. graminis* pathogen fitness. Thus, we hypothesize that in the *Bgh* population the two dominant *CSEP0141* haplotypes, *AVR$_{a10}$* and *AVR$_{a22}$*, have emerged from the action of two opposing selective pressures: sequence conservation that maintains pathogen fitness through retention of effector virulence activity and sequence diversification to escape recognition by MLA10 and MLA22 receptors, respectively.

## Discussion

Previous pathotyping studies of *Bgh* field isolates with barley varieties carrying different powdery mildew *R* genes suggested that the Central European pathogen population can be considered as a single epidemiological unit (*Limpert, 1987*). We have shown here that among 13 *Bgh* isolates from

**Table 1.** *AVR$_{a10}$*/*AVR$_{a22}$* ortholog *BgtE-5921* variants in different *Blumeria graminis* formae speciales.

| | Number of variants in Blumeria graminis f. sp. | | | | |
| --- | --- | --- | --- | --- | --- |
| *AVR$_{a10}$*/*AVR$_{a22}$* ortholog *BgtE-5921* | tritici | dicocci/tritici2 | triticale | secalis | Total |
| Hap_96224 | 29 | | 22 | | 51 |
| Hap_96224 R111G | 2 | | | | 2 |
| Hap_96224 V5I, R111G | 2 | | | | 2 |
| Hap_96224 R111G, A115G | 1 | | | | 1 |
| Hap_96224 F77L, R111G | | 4 | | | 4 |
| Hap_96224 F77L, R111G, Y117H | | 1 | | | 1 |
| Hap_96224 V5I, I8L, F77L, R111G | | 1 | | | 1 |
| Hap_96224 V17I, S47M, D48N, R50K, G58D, G59S, R101C, R111G, A115V | | | | 5 | 5 |
| total number of isolates | 34 | 6 | 22 | 5 | 67 |

DOI: https://doi.org/10.7554/eLife.44471.025



**Figure 6.** Conservation of *AVR_a10/AVR_a22* orthologs between *Blumeria graminis formae speciales.* (**A**) Alignment of protein sequences (AVR_A10/AVR_A22) encoded by *Bgh* CSEP0141 and orthologs detected in *B. graminis* f. sp. *tritici* and *B. graminis* f. sp. *secalis*. (**B**) Number of non-synonymous sequence variants was determined for 190 core effectors (*Frantzeskakis et al., 2018*) among all *Bgh* isolates described in this study and is displayed for all core effectors with ≥1 non-synonymous variants/100 bp coding sequence. Grey bars, including all *Bgh* isolates; black bars, all *Bgh* isolates excluding RACE1.

DOI: https://doi.org/10.7554/eLife.44471.023

The following figure supplement is available for figure 6:

**Figure supplement 1.** Phylogenetic and genomic analysis of *AVR_a7*, *AVR_a9*, *AVR_a10/AVR_a22* homologs in *Blumeria graminis* f. sp. *tritici*.

DOI: https://doi.org/10.7554/eLife.44471.024

a local population in Germany ten virulence combinations can be distinguished in interactions with a panel of *Mla* NILs, suggesting potential adaptation of the pathogen population to multiple *Mla* resistance specificities. Our findings are congruent with a recent study describing a high complexity of Central European *Bgh* virulence pathotypes on a panel of 50 differential barley varieties carrying *Mla* or other powdery mildew *R* genes (*Dreiseitl, 2014*). The same study also provided evidence for a complete separation of the Central European and Australian *Bgh* populations with non-overlapping pathotypes and estimated an almost three-fold higher virulence complexity for the European population. This is believed to be due to the cultivation of barley varieties carrying distinct powdery mildew *R* gene combinations on the two continents, which leads to differential intensities of directed selection on *Bgh* populations. Analysis of our isolate collection comprising, among others, 13 newly characterized *Bgh* strains from a local population in Germany and three representative Australian

isolates, is supportive of a significantly greater virulence complexity of the Central European over the Australian isolates even when considering virulence patterns only on *Mla* NILs (*Supplementary file 1*). The complexity of avirulence and virulence alleles in the Central European population suggests that the fungus adapts by balancing selection of *AVR$_a$* genes among strains rather than selective sweeps to maintain pathogenicity.

Specific activation of cell death upon transient gene expression of matching *AVR$_a$* and *Mla* gene pairs in barley protoplasts and heterologous *N. benthamiana* provided evidence for the function of candidate *AVR$_{a7}$*, *AVR$_{a9}$*, *AVR$_{a10}$*, and *AVR$_{a22}$* as avirulence effectors, all of which encode *Bgh* CSEPs with a predicted signal peptide. This is different from the previously reported EKA_AVR$_{A10}$, which lacks a canonical signal peptide, belongs to the large *EKA* gene family that is derived from part of a class-1 LINE retrotransposon (*Ridout, 2006*; *Amselem et al., 2015*), and was used as an elicitor of induced MLA10 nuclear interaction with a WRKY transcription factor (*Shen et al., 2007*). However, we were unable to detect MLA10-mediated cell death activity for EKA_AVR$_{A10}$. Our findings demonstrate that virulent *Bgh* isolates escape recognition of corresponding MLA receptors predominantly by non-synonymous SNPs but also loss of expression of the corresponding genes (*Figure 3A*; *Lu et al., 2016*). Upon *in silico* removal of the signal peptide, phylogenetic analysis for the 805 predicted secreted proteins of *Bgh* (*Frantzeskakis et al., 2018*) and comparative sequence analysis of *AVR$_{a7}$*, *AVR$_{a9}$*, *AVR$_{a10}$*, *AVR$_{a22}$*, *and* previously reported *AVR$_{a1}$* and *AVR$_{a13}$*, also representing CSEPs, failed to detect evolutionary conservation (*Figure 3—figure supplement 1*) or significant sequence similarity (at most 8% sequence identity) between any pair of these polypeptides except for allelic *AVR$_{a10}$* and *AVR$_{a22}$* (*Figure 3C and E*). By contrast, MLA7, MLA9, MLA10 and MLA13 receptors are >96% identical in aa sequence to each other, whereas MLA1 and MLA22 are more diverged and share 91% identical polypeptide sequences with this receptor group (*Seeholzer et al., 2010*). This raises questions regarding the evolutionary history of MLA10- AVR$_{A10}$ and MLA22-AVR$_{A22}$ receptor-effector pairs. *AVR$_{a10}$* and *AVR$_{a22}$* are notable for several reasons: (i) these allelic effectors define two dominant haplotypes of *CSEP0141* in *Bgh*, (ii) *CSEP0141* belongs to a core of 190 effectors that are conserved among different *B. graminis* f. sp. and, therefore, likely contributes to pathogen fitness, (iii) *CSEP0141* represents the core effector with the highest frequency of non-synonymous SNPs among the tested global collection of *Bgh* isolates, and (iv) a single dominant haplotype of its *Bgt* ortholog, designated *BgtE-5921*, was found in the wheat powdery mildew population. Collectively, this suggests a model in which the two dominant *CSEP0141* haplotypes, *AVR$_{a10}$* and *AVR$_{a22}$*, evolved in *Bgh* by the action of two opposing selective forces: functional conservation that maintains pathogen fitness through retention of effector virulence activity and sequence diversification to escape recognition by MLA10 and MLA22 receptors, respectively. *AVR$_{a10}$* and *AVR$_{a22}$* likely represent a balanced polymorphism in the extant pathogen population because *Bgh* isolates containing one or the other haplotype do not form discrete subgroups (*Figure 1—figure supplement 1A*). However, the likely source of many, if not all, *Mla* recognition specificities in domesticated barley is wild barley, *H. spontaneum* (*Jørgensen and Wolfe, 1994*). Thus, it is possible that *Mla10*, *Mla22* and their matching avirulence effector genes have not diversified during a co-evolutionary arms race (*Ravensdale et al., 2011*) but have rather evolved independently in separate host and pathogen populations. In such a scenario, the balanced AVR$_{A10}$ and AVR$_{A22}$ polymorphism in the extant pathogen population is the consequence of concurrent cultivation of domesticated barley varieties with these *Mla* resistance specificities. Besides an apparently balanced *AVR$_{a10}$* and *AVR$_{a22}$* polymorphism at *CSEP0141* in the *Bgh* field population one would expect sporadic strains that are virulent on both *Mla10*- and *Mla22*-harboring host varieties, which is the case for *Bgh* isolate DH14 carrying a SNP that introduces a single aa substitution in *AVR$_{a10}$* (*Figure 3*). In such strains, a fitness penalty for the pathogen might be mitigated by residual virulence activity of the CSEP0141 variant.

Our split-LUC and yeast two-hybrid experiments provided evidence for direct and specific interactions between multiple matching MLA/AVR$_A$ pairs (*Figure 5*). More than 50 years after the original discovery of multi-allelic race-specific disease resistance at *Mla* (*Moseman and Schaller, 1960*), these findings now imply that the co-evolutionary functional diversification of these immune receptors is at least in part mechanistically underpinned, and was perhaps driven by direct interactions with sequence-unrelated *Bgh* avirulence effectors. Direct receptor-avirulence effector interactions have been described for flax L and rice Pik multi-allelic disease resistance genes, which encode NLR-type receptors, (*Kanzaki et al., 2012*; *Dodds et al., 2006*). In flax, a subset of allelic L receptors (L5, L6, and L7) interact with a specific subset of highly sequence-related flax rust AvrL567 proteins, and

in rice, allelic Pik receptors interact with specific variants of highly sequence-related AVR-Pik proteins. Rice Pik immune receptors contain an integrated heavy metal-associated RATX1/HMA domain, which binds directly to AVR-Pik and *Pik* functional diversification is driven by polymorphisms in this integrated domain (*Kanzaki et al., 2012*; *Maqbool et al., 2015*; *De la Concepcion et al., 2018*). Barley MLA and flax L proteins lack detectable integrated domains, and diversifying selection in allelic MLA receptors is largely confined to predicted solvent-exposed residues of leucine-rich repeats 7 to 15 (*Seeholzer et al., 2010*). Thus, to the best of our knowledge, a co-evolutionary functional diversification of multi-allelic NLR-type receptors in plants with directly recognized sequence-unrelated avirulence effectors, as described here for matching MLA, is without precedence. Effectors AVR1-CO39 and AVR-Pia of the ascomycete *Magnaporthe oryzae* are sequence-unrelated but have very similar 6 β-sandwich structures that are stabilized in both cases by a disulfide bridge and are both recognized by the rice NLR pair RGA4-RGA5 through the integrated RATX1/HMA domain located at the C-terminus of RGA5 (*de Guillen et al., 2015*; *Cesari et al., 2013*). Structural similarity searches then showed that AVR1-CO39 and AVR-Pia are founders of a family of sequence-unrelated but structurally conserved fungal effectors in a broad range of ascomycete phytopathogens (*de Guillen et al., 2015*). Consistent with structural modelling (IntFOLD Version 3.0 (*McGuffin et al., 2015*)), the recently resolved NMR-based and crystal structure of *Bgh* CSEP0064 revealed a ribonuclease-like fold (*Pennington et al., 2019*) with the absence of canonical catalytic residues in the substrate-binding pocket, and the gene products of ~120 additional *Bgh* CSEPs very likely adopt a similar structure (*Pennington et al., 2019*). Structural similarity searches (IntFOLD Version 3.0) also suggested a ribonuclease-like fold for $AVR_{A7}$ and $AVR_{A13}$ (high and certain confidence at p=3.739E-3, and 6.174E-4, respectively), whereas no significant structural similarities were detected between CSEP0064 and $AVR_{A1}$, $AVR_{A9}$, $AVR_{A10}$, or $AVR_{A22}$ (low and medium confidence at p>0.01). Instead, we find that $AVR_{A9}$ may adopt a structural fold that is similar to an antimicrobial peptide called microplusin (p=6.014E-3). In addition, no obvious structural similarities were predicted between $AVR_{A1}$, $AVR_{A9}$, and $AVR_{A10}$/$AVR_{A22}$, suggesting that allelic MLA receptors are capable of detecting the presence of structurally unrelated Bgh effectors. This assumption is consistent with the recent finding that the wheat *Mla* ortholog Sr50 directly binds to the Basidiomycete stem rust (*Pgt*) avirulence effector AvrSr50 (*Chen et al., 2017*). This Basidiomycete effector most likely evolved independently from the Ascomycete *Bgh* effectors and lacks significant sequence and predicted structural similarity with the known $AVR_a$ effectors. We speculate that MLA receptors might have an exceptional propensity to directly detect unrelated pathogen effectors and that this feature might have facilitated the functional diversification of the receptor in the host population.

In a whole leaf context, race-specific disease resistance specified by MLA receptors to *Bgh* is invariably linked to the activation of localized host cell death (*Boyd et al., 1995*). NLR-mediated cell death likely contributes to the termination of biotrophic fungal pathogenesis, including that of powdery mildews, because this class of pathogens feeds on living plant cells. A striking feature of the functional diversification at *Mla* is the enormous variation in microscopic and macroscopic resistance-associated *Bgh* infection phenotypes as shown with barley NILs carrying different *Mla* resistance specificities (*Boyd et al., 1995*). For instance, the onset of detectable host cell death can vary dramatically and can be both rapid and limited to the first attacked leaf epidermal cell, terminating early fungal pathogenesis, or can occur at later stages of fungal pathogenesis and involve numerous leaf mesophyll cells that subtend *Bgh*-infected epidermal cells (*Boyd et al., 1995*). Here we have employed co-transfection experiments of barley leaf protoplasts with *Mla-AVR*ₐ pairs and protoplast cell death as a proxy for receptor activation, excluding the possibility that additional *Bgh*-derived molecules associated with Pattern-triggered immunity influence the timing of immune receptor-mediated cell death in this system. Although based on overexpression data, the significant variation in cell death phenotypes reported here could partly reflect variable *Bgh* infection phenotypes on different MLA NILs (*Supplementary file 1*, *Boyd et al., 1995*). In turn, these differences of infection phenotypes are possibly due to variations in the steady-state levels of the MLA receptors during *Bgh* infection, timing of *Bgh*-mediated $AVR_A$ secretion and/or $AVR_A$ steady-state levels *in planta*, or MLA-$AVR_A$ pair-dependent receptor binding affinities. Whilst establishing the relevance of the the latter requires future biochemical characterization of MLA-$AVR_A$ complexes, our work revealed a very strong binding of $AVR_{A13}$-V2 to MLA13 both in the split-LUC and yeast two-hybrid experiments (*Figure 5*), thereby uncoupling $AVR_A$ binding to the receptor from receptor activation, that is immune receptor-mediated cell death activation. Future biochemical and genetic experiments will clarify

whether the naturally occurring AVR$_{A13}$-V2 effector variant acts as a dominant negative ligand when co-expressed with the AVR$_{A13}$ avirulence effector.

# Materials and methods

**Key resources table**

| Reagent type (species) or resource | Designation | Source or reference | Identifiers | Additional information |
|---|---|---|---|---|
| Strain (*Blumeria graminis f. sp. hordei*) | CC107, CC148, CC1, CC52, CC66, CC88, NCI, 63.5, A6, B103, Aby, Art, Will, OU14, RACE1, K1 | *Lu et al. (2016)* doi:10.1073/pnas.1612947113. | GEO:GSE83237 | |
| Strain (*Blumeria graminis f. sp. hordei*) | DH14 | *Frantzeskakis et al. (2018)* doi:10.1186/s12864-018-4750-6. | GEO:GSE106282 | |
| Strain (*Blumeria graminis f. sp. hordei*) | K2, K3, K4, S11, S15, S16, S19, S20, S21, S22, S23. S25, S26 | this paper | GEO:GSE110266 | collected in 2017 on cv. Meridian and Keeper barley at the Max Planck Institute for Plant Breeding Research, Cologne, Germany (GPS 5°57′N, 6°51′E 5) |
| Recombinant DNA reagent | pIPKb002 | *Himmelbach et al. (2007)* doi:10.1104/pp.107.111575. | NCBI:EU161568.1 | *pZmUBQ:GW, Spc$^R$* |
| Recombinant DNA reagent | pGWB517 | *Nakagawa et al. (2007)* doi:10.1263/jbb.104.34. | NCBI:AB294484.1 | *p35S:GW-4Myc, Spc$^R$* |
| Recombinant DNA reagent | pXCSG-GW-mYFP | *García et al. (2010)* doi:10.1371/journal.ppat.1000970. | NA | *p35S:GW-mYFP, Carb$^R$* |
| Recombinant DNA reagent | pB42AD-GW | *Shen et al. (2007)* doi:10.1126/science.1136372. | NA | *pGal1:B42-AD—1xHA-GW, TRP* |
| Recombinant DNA reagent | pLexA-GW | *Shen et al. (2007)* doi:10.1126/science.1136372. | NA | *pADH1:LexA-BD-GW, HIS3* |
| Recombinant DNA reagent | pDest-GW-nLUC | *Gehl et al. (2011)* doi:10.1111/j.1365-313X.2011.04607.x. | NA | *p35S:GW-Nterminus Luciferase, Kan$^R$* |
| Recombinant DNA reagent | pDest-GW-cLUC | *Gehl et al. (2011)* doi:10.1111/j.1365-313X.2011.04607.x. | NA | *p35S:GW-Cterminus Luciferase, Kan$^R$* |
| Gene (*Blumeria graminis f. sp. hordei*) | AVR$_{a7}$ variants | *Frantzeskakis et al. (2018)* doi:10.1186/s12864-018-4750-6. | csep0059; BLGH_06689; BLGH_06672; BGHR1_17217; BGHR1_17236; BGHR1_17237 | |
| Gene (*Blumeria graminis f. sp. hordei*) | AVR$_{a9}$ variants | *Frantzeskakis et al. (2018)* doi:10.1186/s12864-018-4750-6. | csep0174; BLGH_04994; BGHR1_10042 | |
| Gene (*Blumeria graminis f. sp. hordei*) | AVR$_{a10}$/AVR$_{aa22}$ variants | *Frantzeskakis et al. (2018)* doi:10.1186/s12864-018-4750-6 | csep0141; BLGH_05021; BGHR1_10013 | |

*Continued on next page*

*Continued*

| Reagent type (species) or resource | Designation | Source or reference | Identifiers | Additional information |
|---|---|---|---|---|
| Gene (*Blumeria graminis f. sp. hordei*) | $AVR_{a1}$ variants | *Lu et al. (2016)* doi:10.1073/ pnas.1612947113; *Frantzeskakis et al. (2018)* doi:10.1186/s 12864-018-4750-6 | csep0008; BLGH_03023; BLGH_03022; BGHR1_11142 | |
| Gene (*Blumeria graminis f. sp. hordei*) | $AVR_{a13}$ variants | *Lu et al. (2016)* doi:10.1073/pnas.1612947113; *Frantzeskakis et al. (2018)* doi:10.1186/s 12864-018-4750-6 | csep0372; BLGH_02099; BGHR1_12484 | |
| Gene (*Hordeum vulgare*) | Mla9 | *Seeholzer et al. (2010)* doi.org/10.1094 /MPMI-23-4-0497 | NCBI: GU245941.1 | |
| Gene (*Hordeum vulgare*) | Mla22 | *Seeholzer et al. (2010)* doi.org/10.1094/ MPMI-23-4-0497 | NCBI:GU245946 | |
| Gene (*Hordeum vulgare*) | Mla10 | *Seeholzer et al. (2010)* doi.org/10.1094/ MPMI-23-4-0497 | Mla10 | Different from NCBI:AY266445.1 |
| Gene (*Hordeum vulgare*) | Mla7 | *Seeholzer et al. (2010)* doi.org/10.1094/ MPMI-23-4-0497; *Lu et al. (2016)* doi:10.1186/s 12864-018-4750-6 | Mla7 | Different from NCBI:AY266444.1 |
| Gene (*Hordeum vulgare*) | Mla7 (AAQ55540_ Halterman et al., 2004) | *Halterman and Wise (2004)* doi:10.1111/j. 1365-313X.2004.02032.x | NCBI:AY266444.1 | |
| Gene (*Hordeum vulgare*) | Mla1 | *Seeholzer et al. (2010)* doi.org/10.1094/ MPMI-23-4-0497; *Lu et al. (2016)* | NCBI:GU245961 | |
| Gene (*Hordeum vulgare*) | Mla13 | *Seeholzer et al. (2010) , Lu et al. (2016)* doi:10.1073/ pnas.1612947113. | AF523678.1 | |
| Antibody | monoclonal rat anti-HA | Merck | 3F10, RRID:AB_390914 | 1:2000 |
| Antibody | monoclonal mouse anti-LexA | Santa Cruz Biotechnology | sc7544, RRID:AB_627883 | 1:1000 |
| Antibody | polyclonal rabbit anti-c-myc | Abcam | ab9106, RRID:AB_307014 | 1:5000 |
| Antibody | polyclonal rabbit anti-GFP | Abcam | ab6556, RRID:AB_305564 | 1:5000 |
| Antibody | polyclonal rabbit anti-LUC | Sigma | L0159, RRID:AB_260379 | 1:2000 |
| Antibody | polyclonal goat anti-rat IgG-HRP | Santa Cruz Biotechnology | sc2065, RRID:AB_631756 | 1:100 000 |
| Antibody | polyclonal goat anti-mouse IgG-HRP | Santa Cruz Biotechnology | sc2005, RRID:AB_631736 | 1:100 000 |
| Antibody | polyclonal donkey anti-rabbit IgG-HRP | Santa Cruz Biotechnology | sc-2313, RRID:AB_641181 | 1:100 000 |
| Antibody | monoclonal rabbit anti-GFP | Santa Cruz Biotechnology | sc-8334, RRID:AB_641123 | 1:5000 |

## Plant materials and growth conditions

The barley cultivars (cv.) Golden Promise, Manchuria, and Pallas and their near isogenic lines (*Kolster et al., 1986*; *Moseman, 1972*), were grown at 19°C, 70% relative humidity, and under a 16 h photoperiod. *Nicotiana benthamina* plants were grown in standard greenhouse conditions under a 16 h photoperiod.

## Fungal isolates

Barley leaves suspected of being infected by *Bgh* were collected from the cv. Meridian and Keeper barley fields at the Max Planck Institute for Plant Breeding Research, Cologne, Germany (GPS 5°57′N, 6°51′E 5). Spores of different infected field leaves were transferred onto one-week-old barley leaves of the cv. Manchuria (lacking any *Mla* resistance specificity). Inoculated Manchuria leaves were incubated on 1% Bacto Agar plates supplemented with 1 mM benzimidazole at 20°C, 70% humidity, and long-day conditions for one week until *Bgh* conidiospore growth was visible. Subsequent single spore propagation was applied three (S11, S15, S19, S20, S21, S23, S25, S26) or six (K2, K3, K4, S16 and S22) times for isolation of single *Bgh* isolate genotypes. In total, we collected 13 *Bgh* isolates, which were tested at least three times on a panel of cv. Pallas and cv. Manchuria *Mla* near-isogenic lines. Maintenance of fungal isolates and other *Bgh* isolates in this study was carried out as described previously (*Lu et al., 2016*).

## RNA sequencing

The new RNA-seq data generated for this study are deposited in the National Center for Biotechnology Information Gene Expression Omnibus (GEO) database (accession no. GSE110266). The previously generated RNA-seq data for DH14 and all other isolates can also be found at GEO (accession nos. GSE106282 and GSE83237, respectively).

## RNA-seq read alignment and variant calling

The RNA-seq reads for all datasets were mapped to the new *Bgh* DH14 reference genome assembly (version 4.0) taking into account exon-intron structures using the splice-aware aligner Tophat2 (*Kim et al., 2013*), which considers known splice sites based on the new DH14 gene models (*Frantzeskakis et al., 2018*). Read length was 100 bp for previously sequenced isolates (GSE83237), 150 bp for DH14 (GSE106282), S20 and S25, and 250 bp for all other isolates (GSE110266). To allow for adequate alignment efficiency also for those isolates with higher sequence divergence from the reference genome, we adjusted the alignment settings as follows: –read-mismatches 10 –read-gap-length 10 –read-edit-dist 20 –read-realign-edit-dist 0 –mate-inner-dist 260 –mate-std-dev 260 –min-anchor 5 –splice-mismatches 2 –min-intron-length 30 –max-intron-length 10000 –max-insertion-length 20 –max-deletion-length 20 –num-threads 10 –max-multihits 10 –coverage-search –library-type fr-firststrand –segment-mismatches 3 –min-segment-intron 30 –max-segment-intron 10000 –min-coverage-intron 30 –max-coverage-intron 10000 –b2-very-sensitive. Using the SAMtools suite (Version 0.1.18) (*Li et al., 2009*), the generated alignment files were subsequently filtered to retain only properly paired reads with mapping quality >0 for the downstream analyses.

To assess the expression levels of individual genes, we obtained the fragment counts per gene for each isolate and time point from the mapped RNA-seq reads after filtering using the Subread function 'featureCounts' (version 1.5.0) (*Liao et al., 2014*) with options -t CDS -s 2 M –p. Subsequently, the raw counts were summarized over both time-points for each isolate and normalized to FPKM (fragments per kilobase of transcript per million mapped reads) values for better comparability of expression levels.

In parallel, sequence variants were identified from the mapped RNA-seq reads using two different tools. In both cases, the variant calling was performed on a combined alignment dataset that was obtained by merging the mapped RNA-seq reads from all isolates using the merge function of the SAMtools suite (*Li et al., 2009*). In one approach, single nucleotide polymorphisms (SNPs) were identified using the mpileup function in the SAMtools suite (*Li et al., 2009*) with options -A, -u, -D, -d 30000 –L 7000. The resulting mpileup variants were filtered using SnpSift (Version 3.4) (*Cingolani et al., 2012a*) with filter settings "(AF1 >= 0.01852 and ((DP >= 30) | ((DP >= 10) and (GEN[*].DP>=5))) and (QUAL >= 50) and (GEN[*].GQ>=10) and ((na PV4 | ((PV4[0]>1e-10) and (PV4[3]>1e-5)))" to extract high-quality variants with sufficiently high allele frequency ($\geq$0.01852; that is

alternate allele present in at least ~50% of the reads of one isolate), sufficient read coverage (≥30 reads in total, or ≥10 reads in total and at least one isolate with ≥5 reads), a SNP calling quality score ≥50, at least one isolate with a genotype quality score ≥10, and absence of extreme placement bias. In the other approach, variants were called using freebayes (version 9.9.2) (*Garrison and Marth, 2012*) with options –ploidy 1 –use-duplicate-reads –min-mapping-quality 0 –min-base-quality 20 –min-coverage 30 –genotype-qualities. To allow correct variant calling from our RNA-seq data with freebayes, the mapped reads in this case were preprocessed using the function SplitNCigarReads in the GenomeAnalysis Toolkit (version 3.4.0) (*McKenna et al., 2010*) with options -U ALLOW_N_CIGAR_READS -fixNDN -maxOverhang 10 to split any reads with splice junctions. An additional, independent freebayes variant calling was also performed with –ploidy 2, to allow processing of cases where an isolate contains additional gene copies that differ from each other at some residue(s). The resulting freebayes variant sets were filtered using SnpSift (*Cingolani et al., 2012a*) with filter settings "(AF[*]>=0.01852) and ((exists GEN[*].GQ) and (GEN[*].GQ[*]>=10)) and ((exists GEN[*].AO) and (GEN[*].AO[*]>=3)) and ((DP >= 30) | ((DP >= 10) and (exists GEN[*].DP) and (GEN[*].DP>=5))) and (SAR[*]>0) and (SAF[*]>0) and (RPR[*]>1) and (RPL[*]>1)' to extract high-quality variants fulfilling the same criteria as described above for mpileup. The subsequent variant annotation and effect prediction for all datasets was performed using snpEff (Version 3.4; default settings) (*Cingolani et al., 2012b*) based on the new DH14 genome and gene models.

## Population structure and genetic association analysis of *Bgh* isolates

To obtain a suitable set of single nucleotide polymorphisms (SNPs) for population structure analysis, we further filtered the set of high-quality variants obtained with mpileup from the combined alignment data as described above. We extracted only silent (synonymous) SNPs in coding regions for which exactly two different alleles were found (diallelic SNPs), and complete genotype information was available for all isolates (i.e., no missing data). The resulting set of 6286 high-quality diallelic synonymous SNPs was used to examine the genotype data for the presence of any obvious population structure using the R packages adegenet (Version 2.0.1) (*Jombart and Ahmed, 2011*) and ape (Version 3.4) (*Paradis et al., 2004*). To this end, we created a PCA plot from the genotype data using the function glPca (R package adegenet) and additionally computed a neighbour-joining tree based on the pairwise Euclidean distances between the isolate genotypes using the function nj (R package ape). Additionally, another PCA plot was generated for the European isolates only, based on a set of 5170 high-quality diallelic synonymous SNPs found in the European isolates.

For the association analysis we focused on the high-quality variants obtained with freebayes in the haploid SNP calling, which we filtered further to extract only non-synonymous coding variants predicted to change the protein sequence and generated a simplified genotyping table listing all of these variants. Additionally, we also screened the results from the diploid freebayes variant calling for 'heterozygous' positions with a minor allele frequency of at least 1/3, as in this case the different 'alleles' are likely derived from differing paralog copies, and added these positions to the genotyping table. This procedure was implemented in R (*Supplementary file 2*) and resulted in a set of 22,838 high-confidence variants with predicted effect on the protein sequence, which we tested for their association with the observed avirulence phenotypes using Fisher's exact test.

Loss of avirulence might be caused by different variants in different isolates. Therefore, we integrated all high-confidence non-synonymous variants over each gene to obtain gene-wise genotypes. Moreover, to also include presence/absence polymorphisms we considered the complete absence of a transcript as a 'missing' genotype. Finally, these gene-wise genotypes were tested for association with the observed avirulence phenotypes using Fisher's exact test. This gene-wise integration of variants and the subsequent association test were implemented in R. As further technical validation, we additionally performed the same association test also on gene-wise genotypes obtained in the same way from the high-quality mpileup SNPs (*Supplementary file 3*). All identified $AVR_a$ candidates were picked up using both tools; the p values mentioned are based on the freebayes variants (*Supplementary file 4*).

## Genome sequencing, assembly and annotation

For the improvement of the *Bgh* isolate K1 assembly, genomic DNA was extracted as described (*Feehan et al., 2017*) and sequenced using the Oxford Nanopore MinION platform according to the

manufacturer's instructions for library generation and flow-cell handling. Base-Calling of the resulting long-reads was performed with the Albacore Sequencing Pipeline Software (version 2.0.2, Oxford Nanopore) yielding 3,09 GB of data in 781831 reads, and subsequently assembled with Canu (v1.4, *Koren et al., 2017*). The 2238 assembled contigs were corrected using Illumina short reads (SRR650349 and SRR654727; *Hacquard et al., 2013*) with Pilon (v1.18, *Walker et al., 2014*) in four iterations. Re-annotation of the new genome assembly was performed as described in Frantzeskakis et. al., and CSEPs were then manually curated using WebApollo (v2.0.6, *Lee et al., 2013*). Data are deposited under the accession number PRJEB30373 at EBI-ENA.

## Genome-wide nucleotide diversity of *B.g.* ff. spp. isolates

Illumina short-read sequences of different *B.g.* ff. ssp. (SRA accession number: SRP062198, *Supplementary file 5*) were mapped to the reference genome of isolate 96224 (t ENA accession number: PRJEB28180) as described (*Müller et al., 2019*). For extraction of neutral markers, SNP calling was done with freebayes with default parameters (*Garrison and Marth, 2012*) and SNPs located within genes were excluded (*Müller et al., 2019*). Vcftools (*Danecek et al., 2011*) was used to filter SNPs with the following options: vcf –remove-indels –max-alleles 2 –min-alleles 2 –minDP 8 –maxDP 100 –max-missing 1 –recode –maf 0.01 –minGQ 20. Per-site nucleotide diversity was calculated with vcftools –sites-pi command.

## Maximum likelihood phylogeny for predicted secreted proteins

In order to generate a maximum-likelihood phylogenetic tree based on the mature peptide sequences of the Bgh DH14 SPs, the 805 predicted secreted protein sequences were aligned using MAFFT v7.310 (*Katoh and Standley, 2013*) with the settings –maxiterate 1000 –localpair. The alignment was then passed to IQTree v1.6.beta4 (*Nguyen et al., 2015*) with the settings -nt 10 -mem 12G -bb 1000. The resulting tree was then visualized using iTOL (*Letunic and Bork, 2016*).

## Generation of expression constructs

All genes with or without stop codons, were amplified from the cDNA of *Bgh* isolates (*Lu et al., 2016*) or plasmid templates (*Seeholzer et al., 2010*) using Phusion Hot Start II high-fidelity DNA polymerase (Thermo Scientific) and subsequently cloned into pENTR/D-TOPO ($Km^R$) (Thermo Scientific) or synthesized as pDONR221 ($Km^R$) entry clones from GeneArt (Thermo Scientific). The sequence integrity of all clones was confirmed by Sanger sequencing (Eurofins). Primers for $AVR_a$ amplification were designed to replace the signal peptide with the ATG start codon (*Supplementary file 6*).

For transient gene expression assays *in planta* and for yeast 2-hybrid interaction studies, respective genes were transferred from entry or donor vectors into the expression vectors pIPKb002 ($Spc^R$) (*Himmelbach et al., 2007*), pGWB517 ($Spc^R$) (*Nakagawa et al., 2007*), pXCSG-GW-mYFP ($Carb^R$) (*García et al., 2010*), pLexA-GW ($Carb^R$), or pB42AD-GW ($Carb^R$) (*Shen et al., 2007*) as indicated using LR Clonase II (Thermo Scientific).

For the split-LUC assay, genes of interest were transferred from expression vectors into pDONR207 ($Gm^R$) using BP clonase II (Thermo Scientific) and subsequently cloned into pDEST-GW-nLUC ($Km^R$) or pDEST-GW-cLUC ($Km^R$) (*Gehl et al., 2011*) using LR Clonase II. Alternatively, pENTR/D-TOPO or pDONR221 entry clones were double-digested using PvuI and NruI (NEB) to remove $Km^R$ and linearized constructs were transferred directly into pDEST-GW-nLUC ($Km^R$) or pDEST-GW-cLUC ($Km^R$) using LR clonase II; the integrity of the resulting expression constructs was examined by Sanger sequencing (Eurofins).

## Transient gene expression and cell death assay in barley protoplasts

Assessment of protoplast cell death using a luciferase activity as a proxy for cell viability was adapted from (*Lu et al., 2016*) with the following modifications: cDNAs of the $AVR_a$ candidate genes lacking their respective signal peptides (SPs) were co-expressed together with cDNAs of the corresponding MLA receptors from the same strong ubiquitin promotor in the same barley genetic background to directly compare cell death activities mediated by different MLA and $AVR_A$ pairs. For this, the epidermis of the second leaves from seven to eight-day-old plants of the cultivar Golden Promise was removed before leaves were immersed in the enzyme solution. A total volume of 35 µl water

containing 5 µg of the *luciferase* reporter plasmid, 12 µg of the respective *Mla* construct, and 6 µg of the respective effector construct or an EV was transfected into 300 µL barley protoplasts at a concentration of $3 \times 10^5$ protoplasts/ml solution. For co-transfection of $AVR_{a10}$ and $EKA\_AVR_{a10}$, 5 µg of the *luciferase* reporter plasmid, 10 µg of the *Mla10* construct, 4 µg of $AVR_{a10}$, and either 4 µg of $EKA\ AVR_{a10}$ or 4 µg of EV were transfected. At 16 hr after transfection, protoplasts were collected by centrifugation at $1000 \times g$, the supernatant was discarded, and 200 µl 2x cell culture lysis buffer were added (Promega, E1531). Luciferase activity was determined by mixing 50 µl of protoplast lysate with 50 µl luciferase substrate (Promega, E1501) in a white 96-well plate and light emission was measured 1 s/well using a microplate luminometer (Centro, LB960). Relative luciferase reads (*Figure 4—source datas 1–3*, *Figure 4—figure supplement 1—source datas 1–3*), were calculated by setting the control empty vector sample read of each individual experiment to 1.

## Transient gene expression by Agrobacterium-mediated transformation of *Nicotiana benthamiana* leaves

*Agrobacterium tumefaciens* GV3101::pMP90 and *A. tumefaciens* GV3101::pMP90K were freshly transformed with respective constructs of interest and grown from single colonies in liquid Luria broth medium containing appropriate antibiotics for ~24 hr at 28°C to an $OD_{600}$ not higher than 1.5. Bacterial cells were harvested by centrifugation at $2500 \times g$ for 15 min followed by resuspension in infiltration medium (10 mM MES, pH 5.6, 10 mM $MgCl_2$, and 200 µM acetosyringone) to a final $OD_{600} = 1.2$. Cultures were incubated for 2 to 4 hr at 28°C with 180 rpm shaking before infiltration into leaves from three to five-week-old *N. benthamiana* plants. Bacteria carrying $AVR_a$ constructs or EV plasmid were mixed equally with *Mla* plasmid-carrying bacteria. Tissue for immunodetection analysis was harvested two days post infiltration and cell death scores (*Figure 4—figure supplement 2—source datas 4,5*) were assessed three days post infiltration throughout.

## Plant protein extraction and pull-down for fusion protein detection by immunoblotting

Frozen leaf material was ground to a fine powder using pre-cooled adapters in a bead beater (Retsch) and thawed in cold plant protein extraction buffer (150 mM Tris-HCl, pH 7.5, 150 mM NaCl, 10 mM EDTA, 10% (v/v) glycerol, 5 mM DTT, 2% (v/v) plant protease inhibitor cocktail (Sigma), 1 mM NaF, 1 mM $Na_3VO_4$, 1 mM PMSF, and 0.5% (v/v) IGEPAL) at a ratio of 150 mg fresh tissue/1 ml of extraction buffer. Extracts were centrifuged twice at $15,000 \times g$ for 15 min at 4°C. For SDS-PAGE, extracts were diluted 4:1 with 4x SDS loading buffer and heated to 95°C for 5 min.

For pull-down of mYFP-tagged proteins, GFP-Trap-MA (Chromotek) beads were incubated in equilibration buffer (*Saur et al., 2015*) for 1 hr at 4°C and subsequently mixed with protein extracts for 2 to 3 hr at 4°C with slow but constant rotation. Then, conjugated GFP-Trap beads were washed five times in 1 ml of cold wash buffer (*Saur et al., 2015*) at 4°C before interacting proteins were stripped from the beads by boiling in 25 µl of 4x SDS loading buffer for 5 min.

Samples were separated on 10% SDS-PAGE gels, blotted onto PVDF membrane, and probed with anti-GFP (Santa Cruz Biotechnology sc-8334, RRID:AB_641123; or abcam ab6556, RRID:AB_305564), anti-LUC (Sigma L0159) or anti-c-Myc (abcam ab9106, RRID:AB_307014), followed by anti-rabbit IgG-HRP (Santa Cruz Biotechnology sc-2313, RRID:AB_641181) secondary antibodies. Proteins were detected by the HRP activity on SuperSignal West Femto Maximum Sensitivity Substrate (Thermo Fisher 34095) using a Gel Doc XR +Gel Documentation System (Bio-Rad).

## Split-luciferase complementation assay

To obtain protein extracts for luciferase measurements, one leaf disk with a diameter of 0.38 cm was harvested from three different leaves at two days post transformation, resulting in three leaf disks/sample. Samples were frozen in liquid nitrogen and ground to a fine powder using a pre-cooled adapter in a Retsch bead beater. Each sample was thawed in 100 µl 2x cell culture lysis buffer (Promega, E1531) supplemented with Tris-HCl, pH 7.5 to a final concentration of 150 mM. Luciferase activity was determined by mixing 50 µl of leaf extract with 50 µl luciferase substrate (Promega, E1501) in a white 96-well plate and light emission (*Figure 5—source datas 1–3*) was measured 1 s/well in a microplate luminometer (Centro, LB960). Complementation of a functional LUC protein by genetic fusion of bait/prey with the nucleotides encoding the N-terminal 416 aa of LUC (nLUC) and

C-terminal 152 aa of LUC (cLUC) allows the detection of a real-time and reversible signal for direct interaction (*Bosmans et al., 2016*; *Chen et al., 2008*).

## Yeast 2-hybrid assay and yeast protein extraction

*Mla* variants were cloned into the pLexA-GW vector (*Shen et al., 2007*) for expression with an N-terminal LexA activation domain under the control of a constitutive ADH1 promoter (BD-MLA). The $AVR_a$ variants were cloned into pB42AD-GW (*Shen et al., 2007*) for expression with an N-terminal B42 activation domain followed by the HA-tag under the control of an inducible GAL1 promoter (AD-$AVR_A$). Using the lithium acetate method (*Gietz and Woods, 2002*), *Mla* bait constructs and $AVR_{a13}$ prey constructs were co-transformed into the yeast strain EGY4.8 p8op-*lacZ* and successful transformants were selected by colony growth on SD-UHW/Glu (4% (w/v) Glucose, 0.139% (w/v) yeast synthetic drop-out medium pH 5.8 without uracil, histidine, tryptophan, 0.67% (w/v) BD Difco yeast nitrogen base, 2% (w/v) Bacto Agar). Yeast transformants were grown to $OD_{600} = 1$ in liquid SD-UHW/Glu before harvesting cells for drop out of the dilution series on SD-UHW/Gal/Raf media (SD-UHW without glucose but with 2% (w/v) Galactose 1% (w/v) Raffinose, with (-UHW) or without Leucine (-UHWL)) and incubated for six days at 30°C followed by room temperature incubation for another six days.

For protein detection, yeast strains were grown to $OD_{600} = 1$ in SD-UHW/Gal/Raf liquid medium at 30°C and 200 rpm shaking, and proteins were extracted using 200 mM NaOH (NaOH method; *Zhang et al., 2011*). Total protein samples were separated on 9% or 12% SDS-PAGE gels, blotted onto PVDF membrane, and probed with anti-HA (Merck, clone 3F10, RRID:AB_390914) or anti-LexA (Santa Cruz Biotechnology, sc7544, RRID:AB_627883) primary antibodies followed by anti-rat (Santa Cruz Biotechnology, sc2065, RRID:AB_631756) or anti-mouse IgG-HRP (Santa Cruz Biotechnology, sc2005, RRID:AB_631736) secondary antibodies as appropriate. HA and LexA fusion proteins were detected by HRP activity on SuperSignal West Femto Maximum Sensitivity Substrate (Thermo Fisher 34095) using a Gel Doc XR +Gel Documentation System (Bio-Rad).

## Acknowledgements

We thank Sabine Haigis for maintaining the *Bgh* isolates; Petra Köchner for technical assistance; and the Max Planck Genome Centre Cologne for RNA-seq. This work was supported by the Max-Planck Society (IMLS, SB, BK, XL and PS-L); German Research Foundation in the Collaborative Research Centre Grant SFB670 (to BK, TM, and PS-L); Cluster of Excellence on Plant Sciences (CEPLAS 1028 to PS-L), European Molecular Biology Organization (ALTF 368–2016 to IMLS); Daimler and Benz Foundation (to IMLS) and German Research Foundation-funded Priority Programme SPP1819 (PA 861/14–1 to RP).

## Additional information

### Funding

| Funder | Grant reference number | Author |
| --- | --- | --- |
| Deutsche Forschungsgemeinschaft | SFB670 | Barbara Kracher<br>Takaki Maekawa<br>Paul Schulze-Lefert |
| Max-Planck-Gesellschaft | Open-access funding | Saskia Bauer<br>Paul Schulze-Lefert |
| European Molecular Biology Organization | ALTF 368-2016 | Isabel ML Saur |
| Cluster of Excellence in Plant Sciences | CEPLAS 1028 | Paul Schulze-Lefert |
| Deutsche Forschungsgemeinschaft | SPP1819 | Lamprinos Franzeskakis<br>Ralph Panstruga |
| Daimler und Benz Stiftung | | Isabel ML Saur |

The funders had no role in study design, data collection and interpretation, or the decision to submit the work for publication.

### Author contributions
Isabel ML Saur, Conceptualization, Formal analysis, Funding acquisition, Validation, Investigation, Visualization, Methodology, Writing—original draft, Writing—review and editing; Saskia Bauer, Formal analysis, Validation, Investigation, Visualization, Methodology, Writing—review and editing; Barbara Kracher, Formal analysis, Investigation, Writing—original draft, Writing—review and editing; Xunli Lu, Ralph Panstruga, Investigation, Writing—review and editing; Lamprinos Franzeskakis, Formal analysis, Investigation, Writing—review and editing; Marion C Müller, Formal analysis, Investigation, Visualization, Writing—review and editing; Björn Sabelleck, Florian Kümmel, Methodology; Takaki Maekawa, Formal analysis, Writing—original draft, Writing—review and editing; Paul Schulze-Lefert, Conceptualization, Funding acquisition, Writing—original draft, Project administration, Writing—review and editing

### Author ORCIDs
Isabel ML Saur  http://orcid.org/0000-0002-5610-1260
Saskia Bauer  http://orcid.org/0000-0003-4559-5063
Marion C Müller  http://orcid.org/0000-0001-5594-2319
Ralph Panstruga  https://orcid.org/0000-0002-3756-8957
Paul Schulze-Lefert  http://orcid.org/0000-0002-8978-1717

### Decision letter and Author response
Decision letter https://doi.org/10.7554/eLife.44471.040
Author response https://doi.org/10.7554/eLife.44471.041

## Additional files
### Supplementary files
• Supplementary file 1. Infection phenotypes of *Bgh* isolates on the Pallas and Manchuria cultivar accessions used for the association test.
DOI: https://doi.org/10.7554/eLife.44471.026

• Supplementary file 2. Script for freebayes genetic association analysis.
DOI: https://doi.org/10.7554/eLife.44471.027

• Supplementary file 3. Script for mpileup genetic association analysis.
DOI: https://doi.org/10.7554/eLife.44471.028

• Supplementary file 4. Statistical summary of association analysis.
DOI: https://doi.org/10.7554/eLife.44471.029

• Supplementary file 5. *Blumeria graminis* isolates used for phylogenetic analysis of *CSEP0141* ($AVR_{a10}/AVR_{a22}$).
DOI: https://doi.org/10.7554/eLife.44471.030

• Supplementary file 6. Primers used in this study.
DOI: https://doi.org/10.7554/eLife.44471.031

• Transparent reporting form
DOI: https://doi.org/10.7554/eLife.44471.032

### Data availability
RNA sequencing data have been deposited in GEO under accession code GSE110266 and improved Blumeria graminis f.sp. hordei isolate K1 assembly is deposited under the accession number PRJEB30373 at EBI-ENA. All data generated or analysed during this study are included in the manuscript and supporting files.

The following datasets were generated:

**Database and**

| Author(s) | Year | Dataset title | Dataset URL | Identifier |
|---|---|---|---|---|
| Isabel ML Saur, Saskia Bauer, Barbara Kracher, Lamprinos Franzeskakis, Marion C Müller, Björn Sabelleck, Florian Kümmel, Ralph Panstruga, Paul Schulze-Lefert | 2019 | Improved Blumeria graminis f.sp. hordei isolate K1 assembly | https://www.ebi.ac.uk/ena/data/search?query=PRJEB30373 | European Nucleotide Archive, PRJEB30373 |
| Saur IML, Bauer S, Kracher B, Lu X, Franzeskakis L, Müller MC, Sabellack B, Kümmel F, Panstruga R, Maekawa T, Schulze-Lefert P | 2018 | Identification of the AVRa7,AVRa9, AVRa10 and AVRa22 effector genes from barley powdery mildew fungus (Bgh) association analysis between transcript polymorphisms and AVRa phenotypes from 27 Bgh isolates | https://www.ncbi.nlm.nih.gov/geo/query/acc.cgi?acc=GSE110266 | NCBI Gene Expression Omnibus, GSE110266 |

The following previously published dataset was used:

| Author(s) | Year | Dataset title | Dataset URL | Database and Identifier |
|---|---|---|---|---|
| Lu X, Kracher B, Saur IML, Bauer S, Ellwood S, Wise R, Yaeno T, Maekawa T | 2016 | Identification of avirulence genes in the barley powdery mildew fungus (Bgh) by RNA-sequencing and transcriptome-wide association analysis on a set of Bgh isolates | https://www.ncbi.nlm.nih.gov/geo/query/acc.cgi?link_type=NCBI-GEO&access_num=GSE83237&acc=GSE83237 | NCBI Gene Expression Omnibus, GSE83237 |

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
