## [Decision Letter]

[Editors’ note: a previous version of this study was rejected after peer review, but the authors submitted for reconsideration. The first decision letter after peer review is shown below.]

Thank you for submitting your work entitled "Six pairs of allelic MLA immune receptor-powdery mildew AVR_A_ effectors argue for a direct non-self recognition mechanism" for consideration by *eLife*. Your article has been reviewed by three peer reviewers, and the evaluation has been overseen by a Reviewing Editor and a Senior Editor.

Our decision has been reached after consultation between the reviewers. Based on these discussions and the individual reviews below, we regret to inform you that your work can not be considered for publication in *eLife* in its present form.

The referees strongly acknowledge the comprehensive molecular pathological survey of local *Blumeria graminis* pv. *hordei (Bgh*) isolates and their virulence activities on various barley cultivars, along with the identification and functional verification of numerous novel *Bgh* avirulence factors and their matching barley NLR immune receptors. They also agree that demonstration of physical interaction of NLRs with their corresponding AVRs has major scientific implications as it is contrary to the prevailing view that such events are rare exceptions only. Regretfully, experimental evidence for this latter part is considered not of sufficient quality to support such a major claim. Given that a series of sophisticated methodologies exist to prove and quantify protein-protein interactions in vitro or in vivo, you may choose such techniques to validate the findings obtained by split-luciferase for some of the effector-NLR pairs prior to re-submission of a revised manuscript.

*Reviewer #1:*

This paper reports a comprehensive molecular pathological survey of local *Blumeria graminis* pv. *hordei (Bgh*) isolates and their virulence activities on various barley cultivars. While such surveys have been successfully undertaken in the past, the authors of the present study do not stop by identifying *Bgh*/barley cultivar pairs but identify individual *Bgh* avirulence factors and their matching NLR immune receptors in barley. Functionality of these pairs in plant defense (cell death) activation is tested upon expression in heterologous *N. benthamiana* leaves or in barley protoplasts. The most intriguing finding is demonstration of physical interaction of NLRs with their corresponding AVRs by means of split-luciferase assays in planta (*N. benth.)* and (at least in the case of one combination) in yeast. This is an important finding as it is widely believed that NLR activation is virtually a case of indirect effector or effector activity recognition. Avr-Pita/PiTa interactions (rice/rice blast) are so far the only examples for which such a direct interaction has been demonstrated. In sum, this is a nice molecular survey that is properly conducted. It may not report an exciting novel molecular mechanism but may contribute to a shift in our perception on how NLR-mediated non-self recognition works.

One criticism I have is that biochemical demonstration of ligand-receptor interaction is based upon a rather simple (should I say poor?) set of experiments. Given the technology available to demonstrate (and quantify) receptor ligand binding (affinities), which is also widely used to demonstrate pattern recognition through pattern recognition receptors in plant immunity, the technology used here falls short of what would be possible to demonstrate physical interaction. In particular, reporting affinities between AVR proteins and their corresponding NLRs has not been done before, and would indeed be a true novelty.

*Reviewer #2:*

The manuscript “Six pairs of allelic MLA immune receptor-powdery mildew AVR_A_ effectors argue for a direct non-self recognition mechanism” reports on:

1) The molecular identification of 4 novel barley powdery mildew (*Blumeria graminis* f. sp. *hordei, Bgh*) Avr genes coding all for small secreted proteins (Candidate Secreted Effector Candidate Proteins (CSEPs) by a transcriptome-wide association study (TWAS),

2) The investigation of the molecular details of their recognition by allelic barley *Mla* NLR immune receptors and

3) Analysis of the diversity of the novel Avr_A_ effectors in *Bgh* populations and populations of other formae specialis of Bg.

While part 1 (TWAS-based Avr identification) is quite straightforward and largely validated by transient assays in barley protoplasts and *N. benthamiana* agro-infiltration assays, parts 2 (investigation of association and physical binding between MLA NLRs and *AVR_a_*s) and 3 (diversity analysis) are much less convincing. The major weakness of the study is that direct binding of *AVR_a_*effectors to MLA NLR immune receptors is only weakly supported by the data (in particular, the strong statement in the title is not justified). A minor weakness is that population analysis of the distribution and diversification of the Avr_A_ effectors suffers from lack of description/characterization of the isolates (cf more detailed discussion of the individual parts) and does not bring very interesting new insights. Taken together, the findings in the manuscript are new and of high interest for researchers working on plant immunity and fungal virulence because they broaden knowledge on the molecular identity of fungal effectors recognized by cereal NLR immune receptors and further confirm that highly sequence-conserved *Mla* immune receptors (in certain cases > 96% aa identity) recognize sequence-unrelated effectors.

I recommend to reject the manuscript and to encourage resubmission of a revised manuscript that shows additional data supporting direct binding of *AVR_a_*s to MLAs (and specific binding of recognized *AVR_a_* alleles). Eventually, submission of a strongly revised manuscript that presents and interprets the data on *AVR_a_*/MLA interaction more cautiously would be acceptable. In addition, the description and discussion of the diversity of the novel *AVR_a_*s should be improved.

Specific comments on part 1: TWAS for *AVR_a_* gene identification and validation of candidate genes by transient assays:

This part of the study is straightforward and adds 3 novel *Bgh* Avr_A_ effectors to a recent series of studies that identified 2 *Bgh AVR_a_* effectors (*AVR_a1_* and *AVR_a13_*, Lu et al., 2016), 2 *Bgt* avirulence effectors (NLRs unrelated to *Mla*, Bourras et al., 2015, Praz et al., 2016) as well as 2 wheat stem rust effectors (NLRs highly similar to *Mla*s, Chen et al., 2017 and Salcedo et al., 2017).

Figure 4: The *AVR_a9_* candidate is not validated. Statistical analysis of data from the protoplast assay (panel A, co-expression with *Mla9*) shows no difference to the virulent allele Avr_a9-_V1, Avr_a13-1_ or *AVR_a13_-V2* (all are in class a). Only AVR_a9_-V2 is different. In addition, co-expression with *Mla9* in *N. benthamiana* does not give HR. Based on these data Avr_a9_ cannot considered as validated and paragraph five of subsection “Co-expression of matching *Mla* and *AVR_a_* pairs is necessary and sufficient to trigger cell death in *N. benthamiana*” should be revised accordingly (there is no discrepancy for *AVR_a9_* between protoplast and *N. benthamiana* assay). *AVR_a10_* is not significantly different from *AVR_a10_-V/AVR_a22_-V* in the protoplast assay when co-expressed with *Mla10*. This striking result should be more clearly described. For cell death induction in *N. benthamiana* (panel D of Figure 4), it would be important to have quantitative data since there is high variability in this assay. Best would be use of cell death scoring (4 or five scales based on cell death intensity) and comparison using appropriate statistics and replicate number.

Subsection “Functional analysis of *AVR_a_* candidates in barley leaf protoplasts”: "We detected reduced LUC activity when *AVR_a7_-AUS* was co-expressed with *Mla7*, but statistically this did not differ from protoplasts expressing *AVRa7 -V2* with *Mla7* (Figure 4—figure supplement 1A)". Since the difference between *AVR_a7_*-AUS and *AVRa7 -V2* is statistically not significant you cannot say that LUC activity is reduced. The statistic test says there is no difference!

In the same subsection: There is no statistically significant difference between *AVR_a7_-1* and *AVR_a7_-V1* in Figure 4—figure supplement 1C. Therefore, it does not make sense to insist on a reduction in Luc Activity and the interpretation of the experiment should be accordingly: recognition of *AVR_a7_-1* by MLA7_AAQ55540 is not detected in the protoplast assay.

Specific comments on Part (2) Physical binding between MLA and *AVR_a_* proteins is not convincingly demonstrated by Y2H and split luciferase assay.

Figure 5A, B and C: The split ubiquitin assay shows association between two proteins in planta not direct physical binding. Reconstitution of functional luciferase occurs when nLUC and cLUC are close enough which indicates that the nLUC and cLUC fusion proteins are part of the same protein complex but not necessarily that their interaction is direct. The sentence "Although the LUC complementation assay is suggestive of a direct receptor – avirulence effector interaction, we cannot fully exclude the involvement of other plant proteins in the detected interactions" gives clearly the wrong direction for the interpretation of these experiments.

Figure 5C. It is surprising that the difference between *AVR_a10_* and *AVR_a10_-V/AVR_a22_-V* is statistically not significant because the variance is limited; the statistical test should be verified and if calculation is correct, the number of replicates should be increased. If there is indeed no statistically significant difference between the two constructs such differences should not be claimed. Specific formation of a *Mla22/AVR_a22_*complex is not supported by the split luciferase assay and should not be claimed.

The yeast two hybrid assay in Figure 5D is a key experiment. However, the signal for *AVR_a13_-1*/MLA13 interaction in the Y2H assay is extremely weak (and for *AVR_a13_-3*/MLA13 this is even worse) and not convincing enough for such an important key experiment. Results from quantitative LacZ activity measurements should be provided or other Y2H constructs (in particular GAL4-based constructs) and/or other reporters (in particular His auxotrophy that can be precisely adjusted by varying 3AT concentrations) should be used. It would also be interesting to test the interactions between the other *AVR_a_*s and their corresponding MLAs.

Specific comments on part 3: Investigation of *AVR_a_* effector diversity

Paragraph three of subsection “Evolutionary history of *Bgh AVR_a_* genes and population-level *AVR_a10_ /AVR_a22_* sequence variation in *B. graminis* formae speciales” In the phylogenetic analysis of CSEP0141 using *Bg* isolates other than *Bgh*, it should be indicated which isolates were used, where and when they were sampled and what diversity they are intended to represent. In addition, actual diversity measured with neutral markers should be provided. Otherwise, it is difficult to interpret the information that one single haplotype was identified in f. sp. *triticale* and one in f. sp. *secalis*. Also the frequencies of the different *Bgt* haplotypes is difficult to interpret.

In the same paragraph: It is not correct that there is limited diversity for CSEPS0141 in wheat since 7 haplotypes were detected. The frequencies of these haplotypes in world-wide or European populations remain unclear since there is no documentation on the analyzed *Bgt* isolates.

Comments on the Discussion

Paragraph one: How good is the diversity in Australian *Bgh* populations reflected by the 3 chosen isolates? From neutral diversity (Figure 1—figure supplement 1A) the 3 Australian isolates seem to be extremely similar. Does that reflect the diversity of *Bgh* in Australia? Are this historical or recent isolates? Would additional, well selected isolates add additional neutral and/or pathotype diversity?

Paragraph two: The *AVR_a9_*and the *AVR_a10_* candidates were not really validated by protoplast assays, the *AVR_a9_* candidate was not validated by *N. benthamiana* assays.

Paragraph two: Escape from *Mla*-mediated recognition by loss of expression is not really demonstrated in the study. Only one single isolate lacks expression of the *AVR_a9_*-candidate gene.

“Our split-LUC and yeast-two-hybrid experiments provided evidence for direct and specific interactions between MLA7 and AVR_A7_, MLA10 and AVR_A10_, and MLA13 and AVR_A13_ pairs (Figure 5).”: Sentence incorrect and in insufficiently supported. Y2H was only performed with *AVR_a13_*. Direct interaction is not demonstrated by Split Luciferase.

“The recently resolved NMR based and crystal structure of *Bgh* CSEP0064 revealed a ribonuclease-like fold, lacking canonical catalytic residues in the substrate-binding pocket, and the gene products of ~120 additional *Bgh* CSEPs very likely adopt a similar structure [52].”: This sentence refers to un-published results of another group (cited as submitted manuscript). Data are not available for readers of the manuscript.

“When we used the crystal structure of *Bgh* CSEP0064 as template for structural similarity searches, we identified *AVR_a7_* and *AVR_a13_* as family members (high and certain confidence at p = 3.739E-3, and 6.174E-4, respectively), whereas no significant structural similarities were detected with *AVR_a1_, AVR_a9_, AVR_a10_*, and *AVR_a22_* (low and medium confidence at p > 0.01).”: Sentence refers to data that are not shown. Cannot be verified and properly appreciated.

“Instead, we find that *AVR_a9_* likely adopts a structural fold that is similar to an antimicrobial peptide, called microplusin (p = 6.014E-3).”: Again, data are not shown. In addition, it is unclear what type of modeling was performed since there seem to be no sequence homology and how reliable his modeling is.

“We conclude that MLA receptors might have an exceptional propensity to directly detect unrelated pathogen effectors and that this feature has facilitated the functional diversification of the receptor in the host population”. Highly speculative hypothesis.

“Whilst the latter is subject to future biochemical characterization of MLA – AVR_A_ complexes, our work revealed a very strong binding of AVR_A_13-V2 to MLA13 both in the split-LUC and yeast two-hybrid experiments”. It would be interesting to see hypothesis why there is uncoupling of binding and recognition in the case of *AVR_a13_-V2*/ MLA13.

*Reviewer #3:*

The manuscript "Six pairs of allelic MLA immune receptor-powdery mildew AVR_A_ effectors argue for a direct non-self recognition mechanism" provides new and interesting data on the identification of avirulence genes in the species *Blumeria graminis* and on its potential direct interaction with corresponding NLR-type receptors which are encoded by allelic versions of the R-gene MLA. The manuscript is generally well written and most of the conclusions are justified by the results. It provides new genetic information from a difficult and model system with socioeconomic relevance. Data show an astonishing diversity of a Blumeria population from single local population. It provides evidence for the novel finding that allelic and highly similar MLA receptors can detect non-allelic *AVR_a_*proteins that lack structural conservation. *AVR_a_*proteins likely activate MLA proteins by direct protein-protein interaction. *AVR_a_* genes diversified apparently to avoid recognition and partially balancing selection can observed. I enjoyed reading the manuscript.

The fact that allelic MLA immune receptors and their orthologs apparently detect sequence unrelated fungal avirulence effectors was known before and it was speculated that this is based on direct protein interaction between MLA and *AVR_a_*proteins (Lu et al., 2016). Now the authors newly identified and tested a more comprehensive collection of specific pairs of MLAx and *AVR_a_x* proteins. Here lies novelty and the unique advantage of the system, that authors can test multiple avirulence factors on a series of nearly identical allelic receptors. This allowed for substantiation of previous hypothetical statements. Genetic data appear very solid and overall, I can follow most of the conclusions. However, in quite some details, bioassay/biochemical data are not fully convincing or conclusions are perhaps too strong. I therefore think that the very high potential of this contribution is not yet fully exploited.

I have the following major questions and suggestions:

I am not fully convinced that direct binding potential of the AVR_A_-proteins explains cell death induction and avirulence. Some of your data could be also explained by lack of protein expression or stability. Loss of intrinsic protein stability might be indeed a biologically meaningful and exciting mechanism for avoiding recognition. I think quantification of AVR_A_-protein amounts might help interpreting cell death and split LUC assays more precisely.

I think you should show more positive results for direct protein interaction for at least three of six MLA-AVR_A_ pairs. Show it in vivo instead of protein extracts.

It would be good to show AVR_A_ avirulence function in regard to fungal development by transient expression in epidermal cells. Alternatively, you should reword the manuscript by exchanging avirulence with cell death induction.

[Editors’ note: what now follows is the decision letter after the authors submitted for further consideration.]

Thank you for resubmitting your work entitled "Multiple pairs of allelic MLA immune receptor-powdery mildew AVR_A_ effectors argue for a direct recognition mechanism" for further consideration at *eLife*. Your revised article has been favorably reviewed by three peer reviewers, and the evaluation has been overseen by a Reviewing Editor and Christian Hardtke as the Senior Editor.

The manuscript has been improved but there are some editorial issues that need to be addressed before acceptance, as outlined below:

1) The authors should state explicitly in the Discussion that attempts to produce recombinant proteins for protein-protein interactions studies failed for technical reasons. This will be important to readers unaware of such problems.

2) Your split luciferase assay is performed with protein extracts and not monitored in intact tissue and therefore cannot be considered an in planta assay. This should be reworded.

3)The authors conclude that differences in AVR protein stability is not the dominant mechanism deciding about whether and how strong cell death is executed. However, in single cases you cannot exclude this, and this should be made transparent to the reader to avoid misinterpretation.

4) You need to explain why no protein expression data are provided for barley protoplast assays. Similarly, protein expression data for Figure 5A-C (Figure S4) must be shown in the main figure and explained in the text.

5) It is inappropriate to deduce functional consequences of different natural expression levels of MLA or *AVR_a_* proteins from over-expression data. Here, wording should be more cautious.

6) Summary statistics of the two SNP calling methods must be provided.

7) Your mapping allows up to 10 mismatches per read (subsection “RNA-seq read alignment and variant calling”). Read length and filtering of read lengths are not mentioned. How do you distinguish copy number variants from sequence polymorphism with you methods? Likewise, experimental details on how Pi and which Pi (per gene, per site, per gene per site) were calculated must be provided.

8) Western blots should be shown in the main figures to facilitate interpretation of the cell death, split Luciferase and Y2H results. E.g., the Avr_a13_-V1-nLUC construct that does not give luciferase activity when co-expressed with *Mla13*-cLUC is not detected in WB blot. Therefore, no conclusion can be drawn on the association of Avr_a13_-V1with *Mla13*.

---

## [Author Response]

[Editors’ note: the author responses to the first round of peer review follow.]

The referees strongly acknowledge the comprehensive molecular pathological survey of local Blumeria graminis pv. hordei (Bgh) isolates and their virulence activities on various barley cultivars, along with the identification and functional verification of numerous novel Bgh avirulence factors and their matching barley NLR immune receptors. They also agree that demonstration of physical interaction of NLRs with their corresponding AVRs has major scientific implications as it is contrary to the prevailing view that such events are rare exceptions only. Regretfully, experimental evidence for this latter part is considered not of sufficient quality to support such a major claim. Given that a series of sophisticated methodologies exist to prove and quantify protein-protein interactions in vitro or in vivo, you may choose such techniques to validate the findings obtained by split-luciferase for some of the effector-NLR pairs prior to re-submission of a revised manuscript.

We thank the referees for investing the time to thoroughly evaluate our initial manuscript and for the constructive comments. We agree that our previous submission fell short in validating the findings obtained by split‐luciferase for some of the effector‐NLR pairs, and we are confident that we now address the concerns in this new version and that, in particular, we provide additional convincing evidence for the association of NLRs with their corresponding AVRs.

We have also addressed the other comments of the reviewers and have revised our manuscript considerably.

Reviewer #1:[…] One criticism I have is that biochemical demonstration of ligand-receptor interaction is based upon a rather simple (should I say poor?) set of experiments. Given the technology available to demonstrate (and quantify) receptor ligand binding (affinities), which is also widely used to demonstrate pattern recognition through pattern recognition receptors in plant immunity, the technology used here falls short of what would be possible to demonstrate physical interaction. In particular, reporting affinities between AVR proteins and their corresponding NLRs has not been done before, and would indeed be a true novelty.

We thank the reviewer for these comments and raising these concerns. We agree that our previously submitted manuscript fell short in demonstrating interactions between multiple AVR_A_/MLA pairs. We understand that new technologies are available for protein‐protein interaction studies and would be excited to use these in the future. Unlike surface-resident pattern recognition receptors (PRRs) for which ligand affinities can be obtained in vivo, these methods cannot be applied to date to quantify in vivo ligand binding with labelled ligands for NLRs inside plant cells.

NLRs and their ligand-induced responses are different from PRRs; in particular, MLA-triggered cell death and receptor oligomerisation (Maekawa et al., 2011) means that it has (so far) been impossible to purify large quantities of this receptor (and probably other full‐length NLRs) for in vitro association studies: we have assessed and attempted to test advanced methods for determining protein-protein interactions and for this also collaborate with Jijie Chai, a renowned biochemist and structural biologist with expertise in NLR biology. We considered pull-down assays with recombinant, purified proteins as the most promising approach. However, extensive efforts focused on recombinant expression of sufficient quality AVR_A_s and MLAs in heterologous systems such as insect cells remained unsuccessful. We were forced to conclude that we are at this stage unable to produce MLA and AVR_A_ proteins in sufficient quantity and quality for in vitro protein‐protein interaction assays.

To independently validate the findings obtained by split‐luciferase in planta, we focused to significantly extend the Yeast-2-Hybrid (Y2H) assays. In addition to AVR_A13_-MLA13, we now also demonstrate AVR_A-_MLA interactions that are specific for sequence‐unrelated AVR_A7_ and AVR_A10_ with their cognate MLA receptors in yeast. This corroborates the significance of our original split-LUC effector-receptor association dataset in planta. The new data is now included in Figure 5.

Furthermore, we have obtained evidence for interaction of other AVR_A_ proteins with their cognate MLA in yeast (see in Author response image 1 an example of MLA1 – AVR_A1_ yeast data). However, we are unable to support these yeast data with split‐LUC assays due to very low protein levels of AVR_A1_ and AVR_A22_ in *N. benthamiana* leaves. In addition, co‐expression of *Mla9* and *AVR_a9_* failed to trigger a cell death response in *N. benthamiana* leaves (Figure 4E). For these reasons, we do not wish to include yeast data of these latter MLA – AVR_A_ pairs in the current manuscript.

Reviewer #2:[…] I recommend to reject the manuscript and to encourage resubmission of a revised manuscript that shows additional data supporting direct binding of AVR_a_s to MLAs (and specific binding of recognized AVR_a_ alleles). Eventually, submission of a strongly revised manuscript that presents and interprets the data on AVR_a_/MLA interaction more cautiously would be acceptable. In addition, the description and discussion of the diversity of the novel AVR_a_s should be improved.

We thank the reviewer for sharing his/her concerns. We agree that our previously submitted manuscript fell short in demonstrating direct AVR_A_ ‐ MLA interactions. As requested, we extended the Yeast‐2-Hybrid assays to independently validate the findings obtained by split‐luciferase experiments *in planta*. In addition to AVR_A13_‐MLA13, we now also demonstrate AVR_A_ ‐ MLA interactions that are specific for sequence‐unrelated AVR_A7_ and AVR_A10_ with their cognate MLA receptors in yeast. This corroborates the significance of our original split‐LUC effector‐receptor association dataset in planta. The new data is now included in Figure 5.

In addition, we have also obtained evidence for interaction of other AVR_A_ proteins with cognate MLAs in yeast (see Author response image 1 for MLA1 – AVR_A1_ yeast data). However, we are unable to support these yeast data with split‐LUC assays due to very low protein levels of AVR_A1_ and AVR_A22_ in *N. benthamiana* leaves. In addition, co‐expression of *Mla9* and *AVR_a9_* failed to trigger a cell death response in *N. benthamiana* leaves (Figure 4E). For these reasons, we do not wish to include yeast data of these MLA – AVR_A_ pairs in the current manuscript.

Specific comments on part 1: TWAS for AVRa gene identification and validation of candidate genes by transient assays:This part of the study is straightforward and adds 3 novel Bgh Avr_A_ effectors to a recent series of studies that identified 2 Bgh AVR_a_ effectors (AVR_a1_ and AVR_a13_, Lu et al., 2016), 2 Bgt avirulence effectors (NLRs unrelated to Mla, Bourras et al., 2015, Praz et al., 2016) as well as 2 wheat stem rust effectors (NLRs highly similar to Mlas, Chen et al., 2017 and Salcedo et al., 2017).Figure 4: The AVR_a9_ candidate is not validated. Statistical analysis of data from the protoplast assay (panel A, co-expression with Mla9) shows no difference to the virulent allele Avr_a9_-V1, Avr_a13_-1 or AVR_a13_-V2 (all are in class a). Only AVR_a9_-V2 is different. In addition, co-expression with Mla9 in N. benthamiana does not give HR. Based on these data Avr_a9_ cannot considered as validated and paragraph five of subsection “Co-expression of matching Mla and AVR_a_ pairs is necessary and sufficient to trigger cell death in N. benthamiana” should be revised accordingly (there is no discrepancy for AVR_a9_ between protoplast and N. benthamiana assay). AVR_a10_ is not significantly different from AVR_a10_-V/AVR_a22_-V in the protoplast assay when co-expressed with Mla10. This striking result should be more clearly described. For cell death induction in N. benthamiana (panel D of Figure 4), it would be important to have quantitative data since there is high variability in this assay. Best would be use of cell death scoring (4 or five scales based on cell death intensity) and comparison using appropriate statistics and replicate number.

We thank the reviewer for pointing out this inconsistency. We have carefully checked the raw data for all of our experiments. All the data was generated by two individuals in independent experiments (I. S and S.B.). After re‐inspection of the raw data of all replicates, we realized that the transfection efficiency in half of these individual replicates for the *AVR_a9_/Mla9* dataset was low. This was evidenced by a comparatively poor luciferase activity for the *AVR_a9_* dataset in Figure 4B. As our protocol requires the transfection of three binary plasmids, high transfection efficiency is critical. The replicates with these low‐quality reads have now been repeated according to our standard protocol, resulting in higher transfection efficiency. These new data was now used to generate revised Figure 4B. We speculate that indeed the low transfection efficiency lead to the high variability between samples in Figure 4B, as variability is significantly reduced when including the new high‐quality data.

All *AVR_a10_‐ Mla10* replicates are of high quality and as such, we assume that *AVR_a10_/AVR_a22_‐V* is not significantly different from *AVR_a10_* in this assay, which may be due to the low signal/noise ratio here. Yet, *AVR_a10_* is significantly different from its virulent variant *AVR_a22_* in protoplasts and both *AVR_a10_/AVR_a22_‐V* and *AVR_a22_* do not elicit cell death upon co‐expression of *Mla10* in the *N. benthamiana* leaf assay, for which statistical analysis was now performed (see below).

Regarding cell death induction in *N. benthamiana*

We agree that a more transparent data analysis is favourable for these kinds of assays. We thus scored infiltration symptoms of all replicates. We added graphs and statistical data to Figure 4—figure supplement 2 and mention this in the text and figure legend.

Subsection “Functional analysis of AVR_a_ candidates in barley leaf protoplasts”: "We detected reduced LUC activity when AVR_a7_-AUS was co-expressed with Mla7, but statistically this did not differ from protoplasts expressing AVR_a7_ -V2 with Mla7 (Figure 4—figure supplement 1A)". Since the difference between AVR_a7_-AUS and AVR_a7_ -V2 is statistically not significant you cannot say that LUC activity is reduced. The statistic test says there is no difference!

Indeed, with the statistics performed previously, it remained unclear whether the LUC reduction of *Mla7*+ *AVR_a7_‐AUS* or *AVR_a7_‐V2* is significant. To determine whether the reduced LUC activity of *AVR_a7_‐AUS* and *AVR_a7_‐V2* is specific to *Mla7*, we applied statistical analysis to the combined *Mla1* and *Mla7* dataset. This was possible as the MLA7 and MLA1 samples were always transfected simultaneously.

We now describe the statistical analysis and corresponding *p*‐values in the figure legend (Figure 4—figure supplement 1A). Our data show that *AVR_a7_‐AUS* but not *AVR_a7_‐V2* expression can significantly reduce LUC activity in a *Mla7* but not *Mla1*‐dependent manner, and in the text we have changed the wording to:

“We detected a 30% reduction in LUC activity when *AVR_a7_‐AUS* was co‐expressed with *Mla7* but not when co‐expressed with *Mla1*. Co‐expression of *AVR_a7_‐V2* with *Mla7* did not result in significantly reduced LUC activity when compared to co‐expression with *Mla1* (Figure 4—figure supplement 1A)”

In the same subsection: There is no statistically significant difference between AVR_a7_-1 and AVR_a7_-V1 in Figure 4—figure supplement 1C. Therefore, it does not make sense to insist on a reduction in Luc Activity and the interpretation of the experiment should be accordingly: recognition of AVR_a7_-1 by MLA7_AAQ55540 is not detected in the protoplast assay.

Thank you. This point was also raised by Reviewer #3, who suggested to also perform additional replicates and include these in the data set. We followed this suggestion and performed statistical analysis including the new experimental data and have changed the corresponding *p*‐values in the figure legend.

The additional experiments do not change the overall significance. We thus follow your suggestion and have changed the text to: “co‐expression of *MLA7_AAQ55540* with *AVR_a7_‐2* reduced LUC activity by only 68%. Luciferase activity in protoplasts co‐expressing *MLA7_AAQ55540* and *AVR_a7_‐1* (30% LUC reduction compared to EV) did not differ significantly from protoplasts co‐expressing *MLA7_AAQ55540* and *AVR_a7_V1* (15% LUC reduction compared to EV; Figure 4—figure supplement 1C).*”*

Although *AVR_a7_‐V1* expression reduced LUC activity by 15%, we are unable to determine the relevance of this in terms of infection phenotypes of *AVR_a7_‐V1* carrying *Bgh* isolates.

Specific comments on Part (2) Physical binding between MLA and AVR_a_ proteins is not convincingly demonstrated by Y2H and split luciferase assay.Figure 5A, B and C The split ubiquitin assay shows association between two proteins in planta not direct physical binding. Reconstitution of functional luciferase occurs when nLUC and cLUC are close enough which indicates that the nLUC and cLUC fusion proteins are part of the same protein complex but not necessarily that their interaction is direct. The sentence "Although the LUC complementation assay is suggestive of a direct receptor – avirulence effector interaction, we cannot fully exclude the involvement of other plant proteins in the detected interactions" gives clearly the wrong direction for the interpretation of these experiments.

Thanks, this misleading paragraph was now changed to “Although the LUC complementation assay is suggestive of receptor – avirulence effector associations *in planta*”, and the paragraph now includes the new Y2H‐based AVR_A_/MLA interaction data.

Figure 5C. It is surprising that the difference between AVR_a10_ and AVR_a10_-V/AVR_a22_-V is statistically not significant because the variance is limited; the statistical test should be verified and if calculation is correct, the number of replicates should be increased. If there is indeed no statistically significant difference between the two constructs such differences should not be claimed. Specific formation of a Mla22/AVR_a22_ complex is not supported by the split luciferase assay and should not be claimed.

This is indeed the case; we thank the reviewer for pointing out this issue. As requested by Reviewer #3, we have now also determined the protein levels of constructs expressed to measure luciferase activity in the split‐LUC interaction assay and show representative western blots in Figure S4. At the same time, we also performed additional luciferase measurements as requested and added these data to Figure 5. We performed statistical analysis of all (“old” and “new”) measurements together and have added the data to Figure 5. *p*‐values are indicated in the figure legend.

Based on analysis that takes the additional (new) replicates into account, we indeed found that the interaction of AVR_A10_ with MLA10 was significantly different from that of AVR_A10_/AVR_A22_‐V with MLA10. AVR_A22_/MLA22 is only significantly different from all other constructs tested together with MLA22, although the signal/noise ratio remains low and we mention this in the text. We speculate that this low signal/noise ratio may be due to the comparatively low AVR_A22_ protein levels (new data Figure S4). We do not interpret beyond. Notably, we could not detect EKA_AVR_A10_ protein in any replicate of our association assays and have thus excluded EKA_*AVR_a10_*from these datasets.

The yeast two hybrid assay in Figure 5D is a key experiment. However, the signal for AVR_a13_-1/MLA13 interaction in the Y2H assay is extremely weak (and for AVR_a13_-3/MLA13 this is even worse) and not convincing enough for such an important key experiment. Results from quantitative LacZ activity measurements should be provided or other Y2H constructs (in particular GAL4-based constructs) and/or other reporters (in particular His auxotrophy that can be precisely adjusted by varying 3AT concentrations) should be used. It would also be interesting to test the interactions between the other AVR_a_s and their corresponding MLAs.

Thanks for this suggestion. We now tested for interaction by auxotrophy using a dilution series drop out. Our new data fully corroborate our previous Y2H results (previously *AVR_a13_* only) using the *lacZ* reporter gene, with the advantage that this suggested marker activity can quantitatively determine interaction and provides a clear signal/noise ratio. Thank you for suggesting this significant improvement.

Specific comments on part 3: Investigation of AVR_a_ effector diversityParagraph three of subsection “Evolutionary history of Bgh AVR_a_ genes and population-level AVR_a10_ /AVR_a22_ sequence variation in B. graminis formae speciales” In the phylogenetic analysis of CSEP0141 using Bg isolates other than Bgh, it should be indicated which isolates were used, where and when they were sampled and what diversity they are intended to represent. In addition, actual diversity measured with neutral markers should be provided. Otherwise, it is difficult to interpret the information that one single haplotype was identified in f. sp. triticale and one in f. sp. secalis. Also the frequencies of the different Bgt haplotypes is difficult to interpret.

We thank the reviewer for pointing out this lack of information in our previous manuscript. We have now included the respective information regarding the *Bg* isolates used here in Supplementary file 5 and refer to the file in the text. We have also analysed the genome‐wide nucleotide diversity and found our previous interpretation to be valid. We added the information to the text as follows:

“Based on 1,141 neutral markers, we calculated a genome‐wide nucleotide diversity (π) of 0.022 for the f. sp. *hordei*, 0.013 for the f. sp. *secalis*, 0.050 for the f. sp. *tritici2*, 0.062 for the f. sp. *tritici*, and 0.040 for the f. sp. *triticale* isolates (Supplementary file 5).”

In the same paragraph: It is not correct that there is limited diversity for CSEPS0141 in wheat since 7 haplotypes were detected. The frequencies of these haplotypes in world-wide or European populations remain unclear since there is no documentation on the analyzed Bgt isolates.

Indeed, this information was now added in Supplementary file 5.

Comments on the DiscussionParagraph one: How good is the diversity in Australian Bgh populations reflected by the 3 chosen isolates? From neutral diversity (Figure 1—figure supplement 1A) the 3 Australian isolates seem to be extremely similar. Does that reflect the diversity of Bgh in Australia? Are this historical or recent isolates? Would additional, well selected isolates add additional neutral and/or pathotype diversity?

We thank the reviewer for pointing out this lack of clarity. It has been reported that, in contrast to European *Bgh* populations, the Australian *Bgh* population is characterized by highly similar pathotypes as well as a low genotypic diversity (Kominkova et al., 2016; Dreiseitl et al., 2014; Dreiseitl et al., 2013). In accordance with these observations, for our previous study (Lu et al., 2016), we initially had examined the virulence phenotypes of in total 14 Australian isolates, which, however, did not reveal any further distinct pathotypes. Therefore, we believe that the chosen three isolates, at least for our purposes, are a faithful representation of the Australian *Bgh* population. Accordingly, while an extensive examination of further Australian isolates might have allowed us to incorporate further diversity, the inclusion of further European isolates seemed to be the more promising approach to increase the diversity of our *Bgh* isolate panel to improve the power of our association analysis and facilitate identification of further *Avr* genes.

Paragraph two: The AVR_a9_ and the AVR_a10_ candidates were not really validated by protoplast assays, the AVR_a9_ candidate was not validated by N. benthamiana assays.

Thank you. We have carefully analysed this. Please see our response to your comments on Figure 4 for further details (“Figure 4: The AVR_a9_ candidate is not validated. […]”).

Paragraph two: Escape from Mla-mediated recognition by loss of expression is not really demonstrated in the study. Only one single isolate lacks expression of the AVR_a9_-candidate gene.

Indeed, lack of *AVR_a9_* expression was demonstrated here for *Bgh* isolate CC66, a phenomenon which was also shown previously (lack of *AVR_a1_* expression in *Bgh* isolate NCI, Lu et al., 2016). Although this does not seem to be a frequent event, we cannot ignore these observations and have thus retained our discussion of this possibility in the text: “loss of expression as a way to escape MLA‐mediated recognition”.

“Our split-LUC and yeast-two-hybrid experiments provided evidence for direct and specific interactions between MLA7 and AVR_A7_, MLA10 and AVR_A10_, and MLA13 and AVR_A13_ pairs (Figure 5).”: Sentence incorrect and in insufficiently supported. Y2H was only performed with AVR_a13_. Direct interaction is not demonstrated by Split Luciferase.

Interaction was now exchanged to “association” and as suggested the paragraph and now includes the extended Y2H‐based AVR_A_/MLA interaction data.

“The recently resolved NMR based and crystal structure of Bgh CSEP0064 revealed a ribonuclease-like fold, lacking canonical catalytic residues in the substrate-binding pocket, and the gene products of ~120 additional Bgh CSEPs very likely adopt a similar structure [52].”: This sentence refers to un-published results of another group (cited as submitted manuscript). Data are not available for readers of the manuscript.“When we used the crystal structure of Bgh CSEP0064 as template for structural similarity searches, we identified AVR_a7_ and AVR_a13_ as family members (high and certain confidence at p = 3.739E-3, and 6.174E-4, respectively), whereas no significant structural similarities were detected with AVR_a1_, AVR_a9_, AVR_a10_, and AVR_a22_ (low and medium confidence at p > 0.01).”: Sentence refers to data that are not shown. Cannot be verified and properly appreciated.

We thank the reviewer for pointing out this missing reference. The manuscript is now publicly available at bioRxiv and we added the respective reference.

“Instead, we find that AVR_a9_ likely adopts a structural fold that is similar to an antimicrobial peptide, called microplusin (p = 6.014E-3).”: Again, data are not shown. In addition, it is unclear what type of modeling was performed since there seem to be no sequence homology and how reliable his modeling is.

We thank the reviewer for pointing out this lack of information. We now describe the structural modelling platform (IntFOLD v.3). We further describe in the text why we chose this service (as it is able to predict an RNAse fold for CSEP0064. This was subsequently confirmed by NMR and crystallisation, suggesting a high accuracy of IntFOLD v.3.

“We conclude that MLA receptors might have an exceptional propensity to directly detect unrelated pathogen effectors and that this feature has facilitated the functional diversification of the receptor in the host population”. Highly speculative hypothesis.

We have replaced “We conclude” with “we speculate” and “this feature has facilitated” with “this feature may have facilitated”.

“Whilst the latter is subject to future biochemical characterization of MLA – AVR_A_ complexes, our work revealed a very strong binding of AVR_A13_-V2 to MLA13 both in the split-LUC and yeast two-hybrid experiments”. It would be interesting to see hypothesis why there is uncoupling of binding and recognition in the case of AVR_a13_-V2/ MLA13.

This is indeed an interesting point. Reviewer #3 had a similar suggestion, which we answered in point 32. We believe that this response also clarifies why we do not discuss this important point any further in this manuscript.

Reviewer #3:[…] The fact that allelic MLA immune receptors and their orthologs apparently detect sequence unrelated fungal avirulence effectors was known before and it was speculated that this is based on direct protein interaction between MLA and AVR_a_ proteins (Lu et al., 2016). Now the authors newly identified and tested a more comprehensive collection of specific pairs of MLAx and AVR_a_x proteins. Here lies novelty and the unique advantage of the system, that authors can test multiple avirulence factors on a series of nearly identical allelic receptors. This allowed for substantiation of previous hypothetical statements. Genetic data appear very solid and overall, I can follow most of the conclusions. However, in quite some details, bioassay/biochemical data are not fully convincing or conclusions are perhaps too strong. I therefore think that the very high potential of this contribution is not yet fully exploited.

We thank Rev3 for her/his careful analysis of our data.

Concerning the direct protein interaction between MLA and AVR_A_ proteins (Lu et al., 2016) we would like to clarify that the mentioned publication speculated regarding indirect MLA/AVR_A_ recognition based on dissimilar sequences of AVR_A1_ and AVR_A13_. Here we can now provide biological data to refute this speculation and show that allelic MLAs can directly detect sequence‐unrelated AVRs.

I have the following major questions and suggestions:I am not fully convinced that direct binding potential of the AVR_A_-proteins explains cell death induction and avirulence. Some of your data could be also explained by lack of protein expression or stability. Loss of intrinsic protein stability might be indeed a biologically meaningful and exciting mechanism for avoiding recognition. I think quantification of AVR_A_-protein amounts might help interpreting cell death and split LUC assays more precisely.

Indeed, “loss of protein stability of virulent AVR_a_ variants” was observed for some AVR_A_ alleles, for example AVR_A13_‐V1, AVR_A9_‐V2. In addition to the AVR_A_‐YFP constructs used for *N. benthamiana* cell death assays in Figure S4, we now also show protein levels of constructs used for the split‐LUC assay in Figure S4. (Please also see response to Reviewer 2).

Taken together with previous data (Lu et al., 2016), we conclude that “loss of protein stability” is not the dominant mechanism for “loss of cell‐death inducing function” as the protein levels of the “virulent” variants AVR_A1_‐V1, AVR_A10_‐V, AVR_A9_‐V1 and *AVR_a7_*‐V are similar to those of their respective avirulent variants. Still, these virulent variants are incapable of inducing MLA‐mediated cell death and neither AVR_A10_‐V nor *AVR_a7_*‐V were found to associate with the cognate MLAs in our analyses.

*I think you should show more positive results for direct protein interaction for at least three of six MLA-AVR_A_ pairs. Show it* in vivo *instead of protein extracts.*

We agree that our previously submitted manuscript fell short in demonstrating direct AVR_A_ ‐ MLA interactions. As requested, we extended the Yeast‐2‐Hybrid assays to independently validate the findings obtained by split‐luciferase experiments *in planta*. In addition to AVR_A13_‐MLA13, we now also demonstrate AVR_A_ ‐ MLA interactions that are specific for sequence‐unrelated AVR_A7_ and AVR_A10_ with their cognate MLA receptors in yeast. This corroborates the significance of our original split‐LUC effector-receptor association dataset *in planta*. The new data is now included in Figure 5.

In addition, we have also obtained evidence for interaction of other AVR_A_ proteins with cognate MLAs in yeast (see above example of MLA1 – AVR_A1_ yeast data in response to Rev1). However, we are unable to support these yeast data with split‐LUC assays due to very low protein levels of AVR_A1_ and AVR_A22_ in *N. benthamiana* leaves (Figure S4). In addition, co‐expression of *Mla9* and *AVR_a9_* failed to trigger a cell death response in *N. benthamiana* leaves (Figure 4E). For these reasons, we do not wish to include yeast data of these MLA – AVR_A_ pairs in the current manuscript.

[Editors' note: the author responses to the re-review follow.]

The manuscript has been improved but there are some editorial issues that need to be addressed before acceptance, as outlined below:1) The authors should state explicitly in the discussion that attempts to produce recombinant proteins for protein-protein interactions studies failed for technical reasons. This will be important to readers unaware of such problems.

Thanks. For clarification, we added:

“So far it has been impossible to purify large quantities of recombinant full‐length MLA receptors for in vitroAVR_A_‐MLA association studies, possibly because of MLA‐triggered cell death and receptor oligomerisation (Maekawa et al., 2011). We thus focused on quantitatively measuring putative AVR_A_‐MLA associations in plant extracts using the highly sensitive split‐luciferase (split‐LUC) complementation assay.”

2) Your split luciferase assay is performed with protein extracts and not monitored in intact tissue and therefore cannot be considered an in planta assay. This should be reworded.

Throughout the text, we changed “*in planta*” to “in plant extracts” or deleted *in planta*.

3)The authors conclude that differences in AVR protein stability is not the dominant mechanism deciding about whether and how strong cell death is executed. However, in single cases you cannot exclude this, and this should be made transparent to the reader to avoid misinterpretation.

Thank you, we now mention this in the respective results section. Specifically, we have changed:

“All AVR_A13_ variants except for AVR_A13_‐V1 were detectable in *N. benthamiana* extracts without GFP‐Trap pull‐down.”

to

“AVR_A13_‐1, AVR_A13_‐3, and AVR_A13_‐V2 were detectable in *N. benthamiana* extracts without GFP‐Trap pull‐down (Figure 4E). AVR_A13_‐V1‐mYFP protein was barely detectable even after GFPTrap enrichment (Figure 4E) suggesting that loss of MLA13‐mediated cell death activity for AVR_A13_‐V1 may be due to protein instability”.

4) You need to explain why no protein expression data are provided for barley protoplast assays. Similarly, protein expression data for Figure 5A-C (Figure S4) must be shown in the main figure and explained in the text.

In the results paragraph describing the protoplast-based cell death assay, we now explain the reasoning:

“As epitope tag sequences can interfere with signal‐noise ratios of LUC activity in this assay (Lu et al., 2016), we refrained from fusion of constructs with epitope sequences.”

We understand the necessity of determining protein expression and stability in the heterologous *N. benthamiana* systems and for clarity and direct comparison of phenotype and protein levels, we added the respective western blot analysis to the main figure (Figure 5E; 5F; 5J) as requested.

5) It is inappropriate to deduce functional consequences of different natural expression levels of MLA or AVR_a_ proteins from over-expression data. Here, wording should be more cautious.

We changed the statement to:

“Although based on overexpression data, the significant variation in cell death phenotypes reported here could partly reflect variable *Bgh*infection phenotypes on different MLA NILs (Supplementary File 1, Boyd et al. 1995). In turn, these differences of infection phenotypes are possibly due to variations in the steady‐state levels of the MLA receptors during *Bgh*infection, timing of *Bgh*‐mediated AVR_A_ secretion and/or AVR_A_ steady‐state levels *in planta*, or MLA-AVR_A_ pair‐dependent receptor binding affinities.”

6) Summary statistics of the two SNP calling methods must be provided.

We have added the statistical summary of freebayes and mpileup SNP calling and *p‐values* for significant hits to Supplementary File 4 and refer to this in the results text and Materials and methods section.

7) Your mapping allows up to 10 mismatches per read (subsection “RNA-seq read alignment and variant calling”). Read length and filtering of read lengths are not mentioned. How do you distinguish copy number variants from sequence polymorphism with your methods? Likewise, experimental details on how Pi and which Pi (per gene, per site, per gene per site) were calculated must be provided.

Read length and filtering

We did not apply any filtering other than read length. Read length information of here sequenced *Bgh*isolates is deposited in the NCBI GEO database (accession no. GSE110266). To the respective method section (RNA‐seq read alignment and variant calling) we added:

“Read length was 100 bp for previously sequenced isolates (GSE83237), 150 bp for DH14 (GSE106282), S20 and S25, and 250 bp for all other isolates (GSE110266).”

Copy number variation

We are able to only identify exact gene copy numbers in *Bgh*isolates for which long‐read genome sequences are available (DH14 and RACE1). From short‐read RNA sequencing data alone, copy number variations cannot be reliably identified. The only assumptions one can make from the transcriptome data are on cases where a *Bgh*isolate carries and expresses variable copies of a gene. Such variables appear as “heterozygous” SNPs in transcriptome data indicating the existence of at least two non‐identical copies of this particular gene. These cases are described in subsection “Combined TWAS and *Bgh* genome analysis identified candidates for *AVR_a7_, AVR_a9_*, *AVR_a10_*, and *AVR_a22_*”.

Nucleotide diversity Pi

We mention this information in the Materials and method section (Genome‐wide nucleotide diversity of *B.g.* ff. spp. isolates) and now explicitly state in the Results section that Pi refers to Per‐site nucleotide diversity in this analysis (second paragraph of subsection “Evolutionary history of *Bgh* AVR_a_ genes and population-level *AVR_a10_* /AVR_a22_ sequence variation in B. graminis formae specialis”). The reference (Danecek et al., 2011) for the analytic tool is given in the Materials and methods section.

8) Western blots should be shown in the main figures to facilitate interpretation of the cell death, split Luciferase and Y2H results. E.g., the Avr_a13_-V1-nLUC construct that does not give luciferase activity when co-expressed with Mla13-cLUC is not detected in WB blot. Therefore, no conclusion can be drawn on the association of Avr_a13_-V1with Mla13.

For clarity and direct comparison of cell death and associated phenotypes with protein levels, we added the respective western blot analysis to the main figures (Figure 4E; 4F; 4H and Figure 5E; 5F; 5J) as requested.

We explicitly note in the text that lack of cell death and LUC activity of samples expressing *AVR_a13_‐V1* and *Mla13* is accompanied by undetectable levels of AVR_A13_‐V1 protein in plant extracts.